# Unpacking equity trends and gaps in Nepal's progress on maternal health service utilization: Insights from the most recent Demographic and Health Surveys (2011, 2016 and 2022)

Resham B. Khatri[1,2]*, Rolina Dhital[3], Sabita Tuladhar[4], Nisha Joshi Bhatta[5], Yibeltal Assefa[2]

1 Health Social Science and Development Research Institute, Kathmandu, Nepal, 2 School of Public Health, University of Queensland, Brisbane, Australia, 3 Health Action and Research, Kathmandu, Nepal, 4 Nepal Public Health Association, Kathmandu, Nepal, 5 Family Welfare Division, Ministry of Health and Population, Kathmandu, Nepal

* rkchettri@gmail.com

## Abstract

### Background

Improving maternal health is a global priority for overall socioeconomic development countries, especially in the low- and middle-income countries including Nepal. Recently, Nepal has made significant progress in enhancing access to maternal health services and in reducing maternal mortality ratio (MMR). Nonetheless, the MMR remains high (151 maternal deaths per 100,000 live births), with a slower rate of decline in recent years, particularly among disadvantaged groups. This study investigates trends and determinants of key maternal health services in Nepal.

### Methods

We conducted further analysis of secondary data from the most recent three Nepal Demographic and Health Surveys (NDHS) conducted in 2011 (n = 1,057), 2016 (n = 964), and 2022 (n = 981) among women aged 15–49 who had at least one live birth prior to each survey. The outcome variables for the trend analysis included the uptake of at least four antenatal care (4ANC) visits, institutional deliveries, first postnatal care (PNC) within 48 hours of childbirth, and completion of all these three routine visits. Determinants of institutional delivery, delivery in private health facilities (HFs), cesarian section (CS) deliveries, and uptake of maternity incentive were investigated. Independent variables included socioeconomic characteristics of women and their marginalization status, geographic factors (e.g., province), health system factors (health service use). A multivariable logistic regression analysis was conducted using data from the NDHS 2022 to investigate the associated determinants of outcome variables considering p value <0.05.

**Data availability statement:** All relevant data are within the manuscript and its Supporting Information files.

**Funding:** The author(s) received no specific funding for this work.

**Competing interests:** The authors have declared that no competing interests exist.

## Results

Results showed low completion rates (59%) of all three maternity care visits and significant discontinuity of care throughout the maternity continuum (82% 4ANC, and 73% PNC visits). From 2011 to 2022, there were increased institutional deliveries overall (47% to 81%) and CS within private HFs (30% to 51%), alongside a decreasing trend in the utilization of maternity incentives (87% to 78%). Women from Karnali province and those facing multiple forms of marginalization (women form lower wealth status and who belong to marginalized ethnicities (e.g., Dalits or Janajatis), and lack of education had lower odds of institutional delivery. Conversely, women who attended at least 4ANC visits had higher odds of institutional delivery. Higher odds of childbirth in private HFs were identified in the Koshi, Bagmati, Madhesh, and Lumbini provinces, particularly among women with fewer forms of marginalization. In contrast, women who worked as manual labor or those with higher birth orders had lower odds of childbirth in private HFs. Notably, higher odds of delivery by CS were observed among older women, women who were Maithili native speakers, and in provinces where higher delivery in HFs. Furthermore, the odds of uptake of maternity incentives were lower among women who had gave births in private HFs.

## Conclusions

Marginalized women experience lower uptake of routine maternity care visits and higher discontinuation along the antenatal through to ponstantal period, creating significant equity gaps in Nepal. The increasing trend of deliveries in private HFs, particularly deliveries by elective CS without maternity incentives could lead to financial hardship while seeking routine maternal health care. Health systems should adopt targeted strategies addressing specific needs, considering intersecting marginalization factors. Key interventions include improving infrastructure, hiring and training local health workers, revising maternity incentives, regulating private HFs, and conducting quality audits, including increasing trends of CS deliveries.

## Introduction

Maternal health remains a critical global public health challenge, with an estimated 260,000 maternal deaths annually. The global maternal mortality ratio (MMR) was reduced by about 40% between 2000 and 2023 but still stood at approximately 152 per 100,000 live births [1]. South Asia, including Nepal, accounted for around 17% of global maternal deaths, with a regional MMR around 113 per 100,000 live births, reflecting persistent disparities [2]. Low- and middle-income countries bear over 90% of these deaths, often due to preventable causes during pregnancy and childbirth [2]. Institutional delivery rates have significantly increased over recent decades, especially in South and Southeast Asia, driven by policies like the Sustainable Development Goals (SDGs) promoting facility births to reduce maternal mortality

[3]. Furthermore, cesarean section (CS) rates have also risen, reflecting changes in clinical practice but raising concerns about appropriate use [3]. Maternity incentive programs, such as financial support for facility deliveries, have effectively encouraged women to seek skilled birth attendance, notably improving institutional deliveries in low- and middle-income countries [4]. These combined trends contribute to better maternal outcomes but pose challenges in ensuring equitable access and avoiding unnecessary CS [5]. Although overall, uptake of maternal health services has increased over the last two decades, progress among the most disadvantaged women's groups is slow [6–8]. The maternal mortality ratio (MMR), and neonatal mortality rate (NMR) are still high in Nepal. For instance, the maternal mortality study reported MMR of 151 per 100,000 live births in Nepal in 2021 but still faces challenges to meet the SDG target of under 70 by 2030 [9], while NMR has remained stagnant from 2016 to 2022 (21 per 1000 live births) [10,11]. Higher maternal deaths occurred among women with no or low levels of education [12]. Similar equity gaps exist in NMR, with the national average at 21, the lowest wealth quintile, and illiterate women having an NMR of 31, compared to 13 for households with the highest wealth quintile [11].

Over the past few decades, Nepal witnessed an increase in household income, wealth status, life expectancy, access to education, and basic health services (BHS) [13]. These improvements in socioeconomic status and access to services have enhanced health indicators. However, considerable disparities persist in wealth, access to better education, and health outcomes across the country. Wealth disparities significantly impact health equity by restricting access to health care services for individuals with lower income, resulting in poorer health outcomes [14].

The equity gaps extend beyond the maternal and newborn health (MNH) outcomes and are also reflected in the uptake of routine MNH services in Nepal. The institutional delivery rate among women from the lowest wealth quintiles increased from 4% to 66% from 2006 to 2022; however, the growth is less impressive compared to the rise from 55% to 98% among the richest wealth quintiles during the same period [11,15]. Moreover, the uptake of the institutional deliveries among disadvantaged ethnic groups, such as Dalits (70%) and Madhesi (76%), is much lower than their privileged counterparts, such as Brahmin/Chhetri (87%) [11]. Provincial inequities in health services are quite prevalent as well, with a lower institutional delivery rate among women living in Madhesh (67%) compared to women living in Sudurpaschim province (87%) in 2022 [11]. Our previous analysis also showed that the contact coverage of routine MNH visits was low among women with multiple forms of socioeconomic disadvantages (e.g., triple forms of disadvantage, i.e., women who were from the lower wealth quintiles and marginalized ethnicities such as Dalits and disadvantaged Janajatis, and illiterate) compared to women with two or more forms of privileges (higher wealth status or literate or from Brahmin/Chhetri/Newar ethnic groups) [16].

The Government of Nepal (GON) has prioritized health equity in its policy and programs, aligning with global initiatives such as the Millennium Development Goals (2000–2015) and health-related Sustainable Development Goals (SDG 3) by 2030 [17]. The National Health Policy 2019, aligned with constitutional mandates, aims to strengthen social health protection and ensure access to and use of quality health services [18]. In line with these global and national policies on health, maternal health care services including care for pregnancy and childbirth are available at the grass-root level health facilities (HFs). For instance, antenatal care (ANC) services are available up to the health post levels in each ward (lowest administrative units) and childbirth services are available up to the birthing center level (selected health posts equipped with skilled birth attendants for child birth services) [19]. However, there have been emerging challenges such as increasing the unregulated private health services and lack of health insurance leading to overmedicalization of routine maternal health services (increased elective CS rates), where clients need to pay for the services that are available free of cost in the public HFs, and inadequate provision of equitable and quality health services [20,21].

Disparities also exist in the type and quality of health services available in private and public HFs in Nepal. While access to free BHS in the HFs is the constitutional right of citizens [22], free-of-cost BHS are only available in public HFs through government funding. In contrast, individuals must pay for services in private HFs, even for BHS. Except for a few private hospitals that are enrolled on the Maternity Incentive program, which provides free maternal health services, and the GON reimburses unit cost. Nevertheless, many women prefer private HFs over public HFs due to the perceived better

quality of care [23]. This preference persists despite the concerns about the profit motive, driving over-medicalized care and poor regulated private HFs [24].

Meanwhile, the readiness of public HFs for MNH services remains suboptimal and inadequate compared to private HFs [25]. Although the country has transitioned from a centralized to a federalized health system in 2015, introducing more locally relevant programs and a Gender Equality and Social Inclusion (GESI) approach for equitable policies and strategies including in health sector, its implementation remains limited, leaving some population groups behind in accessing health services [26]. Additionally, the Aama Suraksha program (maternity incentive program) has been implemented since 2005 to provide financial support as transportation costs for those who complete at least four ANC visits as per protocol (transport incentive of NPR 3,000 (mountain), NPR 2,000 (hill), and NPR 1000 (Terai)) and free childbirth services at HFs [27–29]. This program is a promising initiative to improve maternal health in the country by incentivizing pregnant women to use HFs to give birth rather than giving birth at home and risking complications [27,29]. In addition, the Aama Suraksha program ensures free delivery services, normal, instrumental and CS, to women, while BHS only provides free normal delivery services [30,31].

Despite these efforts, and freely available MNH services in Nepal, not all women are seeking care during pregnancy and childbirth in Nepal. There is a lack of evidence on the trends of routine maternal health service utilization especially women of multiple forms of marginalization status. In the context of increased institutional delivery, and increased CS rates especially in private HFs, there have been limited investigation into who are undergoing private HFs for childbirth and CS. Earlier evidence suggest that maternity care program is one of the most equitable program [32], but evidence are lacking on who are benefiting from the maternity incentive program in the recent survey.

Nepal has committed to reduce the MMR from 281 per 100,000 live births in 2006–116 by 2022 and 99 by 2025, and 70 by 2030 [9]. To achieve this target, Nepal needs to achieve universal coverage of quality MNH services, especially those population groups with high MMR and address health inequity more effectively. It is imperative to investigate the extent of inequities so that strategies and programs can be designed to ensure equitable access to MNH services for the populations most in need [28]. The importance of inequities is vital to formulate and revise policies, strategies, and programs in the context of the federalized health system of Nepal. Comprehensive evidence of trends in inequities and associated determinants could inform decision-makers in designing policies and implementing strategies for the most disadvantaged (e.g., multiple forms of marginalization) women. Therefore, this study aims to investigate the equity gaps in key maternal health services, including institutional delivery, place of institutional delivery, delivery by CS, and uptake of maternity incentives in Nepal. To gain a comprehensive understanding of the trends, this study specifically examines the overall trends of key maternal services in the three most recent Nepal Demographic and Health Surveys (NDHSs) from 2011, 2016, and 2022, while also investigating the levels and determinants of use of key maternal health services in NDHS 2022.

## Methods

### Study design, settings, participants

This study is further analysis of publicly available cross sectional data derived from the most recent three nationally representative NDHS data conducted in 2011 [33], 2016 [10], and 2022 [11]. To enhance the manuscript quality, we included most of the components of Strengthening the reporting of observational studies in epidemiology statement [34]. We obtained this data from the Demographic and Health Survey (DHS) Program (www.dhsprogram.com) with their approval and authorization. Participants' details and sampling methodology are described in the NDHS reports [10,11,33]. In brief, the NDHS adopts a two-stage cluster sampling design, with probability proportional to size (PPS). The PPS sampling design is commonly used by LMICs conducting nationally representative surveys. The cluster PPS sampling design captures representative samples from a geographically and ethnolinguistically diverse country context, such as Nepal. In the first stage of NDHS, each province was stratified into urban (further divided proportionate to population size) and rural

areas, called as enumeration areas (EAs). Wards are primary sampling units (PSUs or clusters) selected independently. In the second stage of the NDHS, a total of 30 households per EA were selected with an equal probability of systematic selection from the household listing. The NDHS sampling weights have been calculated and applied, so results are representative at the national as well as strata levels. Each NDHS collects information on pregnancy, childbirth and postnatal care from women who had a live birth in the five years preceding the survey. However, this study included women aged 15–49 years who had a live birth in the one year preceding the survey.

## Sample included in the analysis

We conducted an overall national trends analysis of key maternal health services using data of the most recent three NDHSs (2011, 2016, and 2022). Determinants of key maternal health services were identified using data from the NDHS 2022. The sample included for trend analysis were women aged 15–49 years who had a live birth one year prior to NDHS 2011 (n = 1,057), NDHS 2016 (n = 964), and NDHS 2022 (n = 981) [28]. We used NDHS 2022 data to investigate the determinants of institutional delivery (N = 981) and determinants of institutional delivery in private HFs (N = 796), delivery by CS (N = 796) and uptake of maternity incentive (N = 796).

## Study variables

**Outcome variables.** Ten outcome variables were included in the trends analysis of maternal health services (at least 4 ANC visits, institutional delivery; PNC visit within 48 hours of childbirth; completion of 4 ANC visits and institutional delivery and PNC visits; place of delivery; birth in private HFs; delivery by CS; CS by place of delivery; uptake of maternal incentives; and uptake of maternity incentive by place of delivery). Four outcome variables were considered to investigate the determinants of maternal service utilization (institutional delivery, types of HF births (public or private HFs), delivery by CS, and uptake of maternity incentives).

**Independent variables.** Independent variables included maternal age, religion, ethnicity, education, wealth quintile, marginalization status, province, place of residence, ecological region, occupation, native language, birth order of the child, place of delivery, and uptake of at least 4 ANC visits. The detailed categorization of each independent variable is presented in S1 Appendix. The variable "marginalization status" were created using three background variables: education, wealth status, and ethnicity [16,35,36]. GON has categorized ethnicities into broader six categories (Dalit, Muslims, Terai caste, disadvantaged Janajatis, Brahmin/Chhetri, advantaged Janajatis). In this study, we grouped those six ethnic categories into two groups according to their comparative privileges: socially excluded ethnicities (including Dalit, Muslims, Terai caste, and disadvantaged Janajatis) and advantaged ethnicities (including Brahmin/Chhetri and advantaged Janajatis). Similarly, education status was dichotomized into illiterate (those who cannot read and write) and literate (those who can read and write and who have primary education or higher). In the NDHS, wealth quintiles were constructed (using principal component analysis (PCA)) based on more than 40 asset items being owned by households. Wealth quintiles were dichotomized into two groups, merging the lowest two quintiles as lower wealth status (lower 40%) and the upper three quintiles as upper wealth status (upper 60%). In the second step, by combining three variables with two categories, we created a new variable (marginalization status) with eight categories. We merged three categories with at least one form of marginalization into one category, and the same was done for two forms of marginalization in the next category. So, the new variable with multiple forms of marginalization status had four categories: triple, double, single, and no marginalization [16,28].

## Conceptual framework of study

As a guiding framework, we adapted and modified the framework from Marmot [37,38] and the WHO Social Determinants of Health framework [16] [Fig 1]. Health equity is shaped by structural, intermediary, and health system factors.

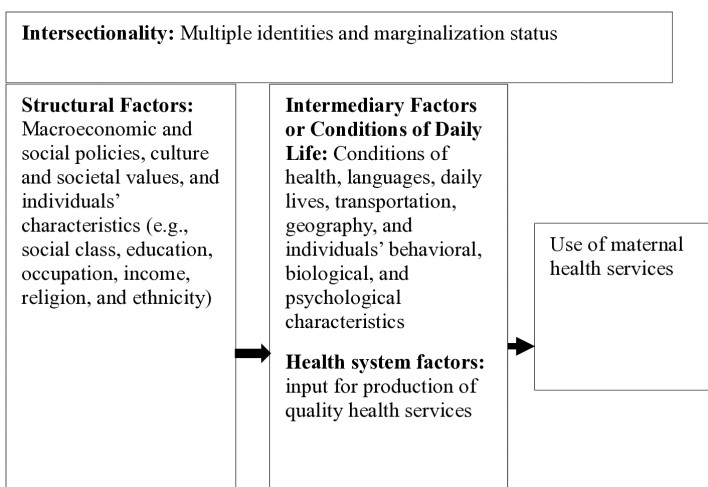

**Fig 1. Conceptual framework to guide the study.**

Structural inequities like education, ethnicity, wealth, and multiple forms of marginalization require political reforms, as these are beyond health system reach. Intermediary factors, such as living and working conditions and infrastructure, are modified through development efforts to improve access. Health systems influence equity by enhancing service readiness and quality of care. Furthermore, individuals experience both structural factors (rooted in societal structures and requiring political interventions) and intermediary factors (modifiable through multisectoral efforts), which can influence towards equity of health and social services. These interconnected factors intersect across multiple identities, creating layered marginalization that affects access to maternal health services. Addressing all three levels is essential to effectively improve health equity. Based on this framework, this study selected independent variables including intersectional marginalization, structural factors (religion, ethnicity, wealth, occupation), and intermediary factors (maternal age, province, ecological region, residence, birth order, native language) and health system factors (uptake of least 4ANC visits).

## Data analysis

Analysis was conducted with two different approaches for trends analysis and determinants of outcome variables. We conducted descriptive analysis of trends and reported the proportion in the Figures and conducted test of significance of trends using proportion test (Stata command "prtesti"). To identify the associated determinants with several maternal health service utilization, we performed univariable (distribution of sample), bivariable (association of each independent variable with each outcome variable), and multivariable (adjusted model) logistic regression analysis. Before running the final regression model, the study checked multicollinearity and excluded independent variable (eco-region was excluded) with variance inflation factor (VIF) ≥5. The variable on multiple marginalization status was created using wealth status, ethnicity, and education, and was collinear with three variables. So, we excluded wealth status, ethnicity, and education in the adjusted regression model. The study reported the adjusted odds ratios (AOR) with a 95% confidence interval (CI), and the statistical significance level was set at $p < 0.05$ (two-tailed). All analysis output included in this study are weighed estimates (otherwise indicated). All analyses were conducted using the "svy" command function which adjust clustering effect of survey design by adjusted cluster variable along with stratification and sampling weights and ensure correct variance estimation and inference. All analyses were conducted in Stata 17 (Stata Corp, 2023).

### Research ethics

We used publicly accessible, de-identified datasets from the Demographic and Health Survey program ([https://dhspro-gram.com/data/available-datasets.cfm](https://dhspro-gram.com/data/available-datasets.cfm)). The procedure for the ethical clearance of NDHSs are written in the original reports. Given that this research involved only a secondary analysis of completely anonymised data, there was no need for additional research ethics clearance. The author team received approval to download and utilize the de-identified data for the further analysis of NDHSs report and for this analysis.

## Results

### Trends of ANC, institutional delivery and PNC (2011–2022)

Fig 2 presents trends in at least 4ANC visits, institutional delivery, at least one PNC within the first 48 hours of childbirth by marginalization status [Supplementary information contains detailed information (S1 Appendix)]. At the national level, the overall trends were increasing for all routine maternal services from 2011 to 2022 for all study groups. For instance, 4ANC uptake among women with triple forms of marginalization increased from 27% in 2011 to 50% in 2016 and to 67% in 2022. However, the uptake of each type of maternal service showed a decline from 4 ANC to PNC for all groups. For instance, in the NDHS 2022, women with triple forms of marginalization had 82% 4 ANC uptake, and institutional delivery (81%), and at least one PNC visit (73%), but uptake of all routine care visits was only 59%.

Uptake of all routine maternal health services was the highest among women with no disadvantages and the lowest among women with triple forms of marginalization/disadvantage. The equity gap between all four marginalization status categories declined over time. From 2011 to 2022, there was a statistically significant increase in the uptake of at least 4 ANC visits for all women with at least one marginalization status (66% in 2011, 75% in 2016, and 86% in 2022). The increase in institutional deliveries and PNC visits from 2011 to 2022 was statistically significant for all categories of marginalization status. The proportion of women who have completed all three routine maternal care visits (4 ANC, HF delivery and PNC) increased significantly from 2011 to 2022 for all women with at least one form of marginalization.

### Trends of place of delivery

Fig 3 illustrates the trend of place of delivery (among all women who had at least one live birth one year prior to each survey) over the three NDHSs. The childbirth trends at both public and private HFs from 2011 to 2022. For instance, childbirth at public HFs increased from 36% to 63% from 2011 to 2022, while at private HFs, it increased from 11% to 18% in the same duration. Despite the decline in home delivery without assistance over the years, they still accounted for 19% of all women who gave birth, as reported in NDHS 2022.

### Trends of delivery by CS and uptake of maternity incentive

Fig 4 illustrates the trends of delivery by CS and the uptake of maternity incentives among women who had at least a live birth and gave birth in HFs (among all women who had institutional delivery). The national trend for all women in CS rates nearly doubled from 13% in 2011 to 25% in 2022. Delivery by CS was generally highest among women, with no disadvantage or a single form of disadvantage. Among those women who gave birth in HFs (both public and private), the uptake of maternity incentives decreased from 73% in 2011 to 67% in 2022.

### Trends of delivery by CS and Trends of the uptake of maternity incentives in public and private HFs

Fig 5 shows the trend of delivery by CS and the uptake of maternity incentives in public and private HFs in the three NDHSs (among all women who had institutional delivery). From 2011 to 2022, the rate of delivery by CS in public HFs increased from 9% to 18%, while in private HFs, it increased from 30% to 51% in the same period. Similarly, the uptake of maternity incentives by public and private HFs (not all private HFs had the Aama program) showed births in three NDHSs.

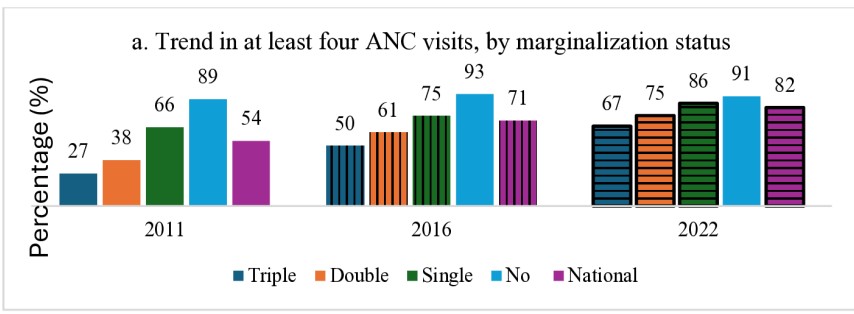

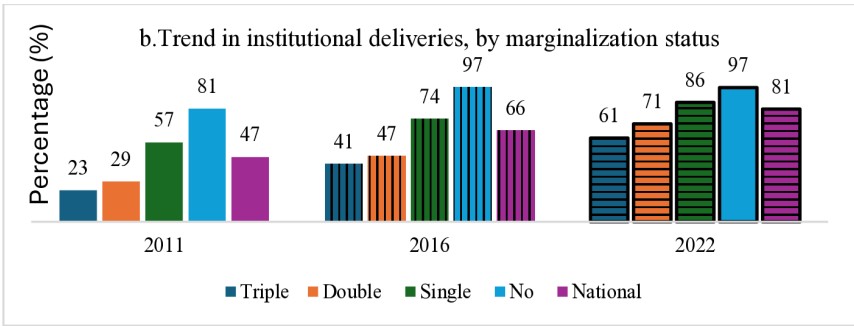

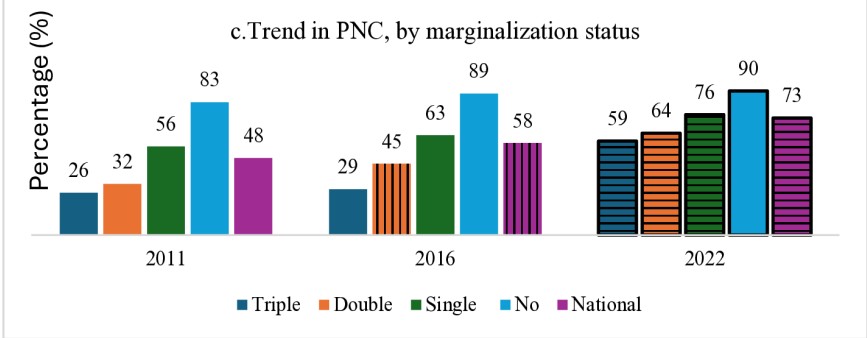

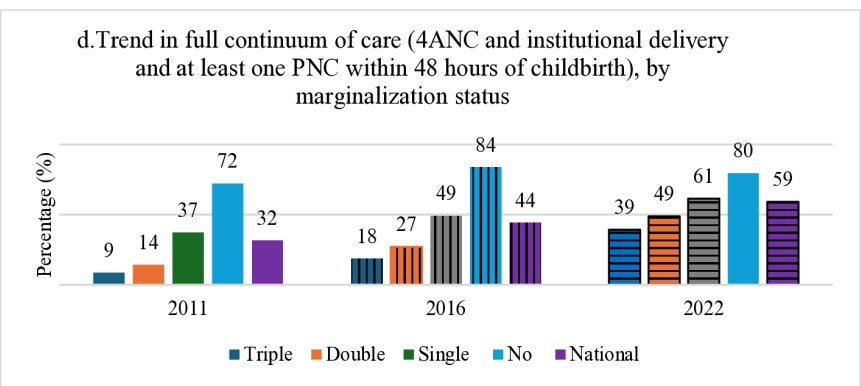

Note: Cut off p value for significance test (p<0.05). Each bar with vertical lines indicates a statistically significant change from 2011 to 2016. Each bar with horizontal lines indicates a statistically significant change from 2016 to 2022. Each bar with a solid outline indicates a statistically significant change from 2011 to 2022.

**Fig 2. Trends of uptake of routine maternal health services among women who had a live birth one year prior to the survey in the three most recent surveys, by marginalization status.**

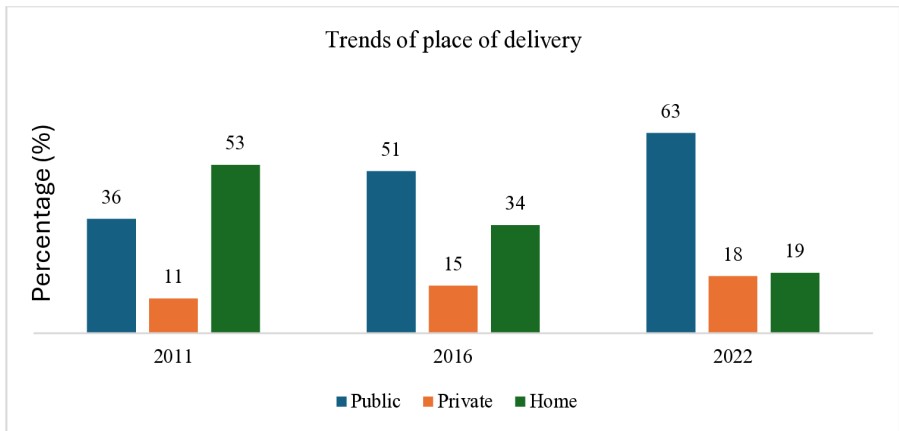

**Fig 3. Trends of place of delivery among women who had at least one live birth one year prior to the survey in the three most recent surveys.**

Among women who gave birth in a public HF, the uptake of maternity incentives showed a declining trend. Nearly 9 in 10 (87%) women received maternity incentives in 2011, which declined to 78% in 2022. Among those who gave birth in private HFs, the proportion of women receiving maternity incentives has remained low at 29% in 2011 and 2022, slightly increasing to 40% in 2016.

### Descriptive analysis of the women who had live birth at least one year prior to the survey, NDHS 2022)

Table 1 presents the background characteristics of the study participants. In NDHS 2022, a high proportion of women were aged 20–24 years (36.8%), from Janajati backgrounds (31.9%), with secondary education (44.9%), from Madhesh province (24.4%), with their first birth (40.7%), and with at least one form of disadvantage (43.4%).

### Descriptive analysis of institutional delivery, delivery in private HF, delivery by CS, and uptake of maternity incentives in NDHS 2022

Detailed description on the uptake of institutional delivery by study variables, and their bivariable association (chi-square association) were included in S1 Appendix. Briefly, among 981 study participants, 796 (81.1%) delivered their babies at HFs. Nine background variables were significantly associated with institutional delivery. Large equity gaps existed between Brahmins (90.8%) and Muslims (68.7%), women with no education (64.3%) and those with higher education (100%), women in the highest wealth quintile (96.7%) and those in the lowest (71.6%), women with triple disadvantages (60.2%) and women with no disadvantages (96.6%), women in Madhesh province (64.8%) and those in Bagmati province (94.4%), Maithili speakers (64.1%) and Nepali speakers (91.4%), and women with a third or higher birth order (63.5%) and those with a first birth order (91.3%). Detailed description on the uptake of institutional delivery by study variables, and their bivariable association (chi-square p values) were included in the.

Among all institutional deliveries, delivery in private HFs by study variables, and their bivariable association (chi-square association) were included in S1 Appendix. Among all institutional deliveries (n = 796), more than one in five women (21.6%) gave birth in private HF. Seven background variables were significantly associated with delivery in a private HF. Compared with the national average and reference categories for each variable, delivery at private HF was high among Muslim women (41.2%), women with higher education (42.5%), women in the highest wealth quintile (31.8%), women from Koshi province (35.7%), women from the Terai ecoregion (27.4%), women with paid jobs (29.4%), and Maithili speakers (38.5%)..

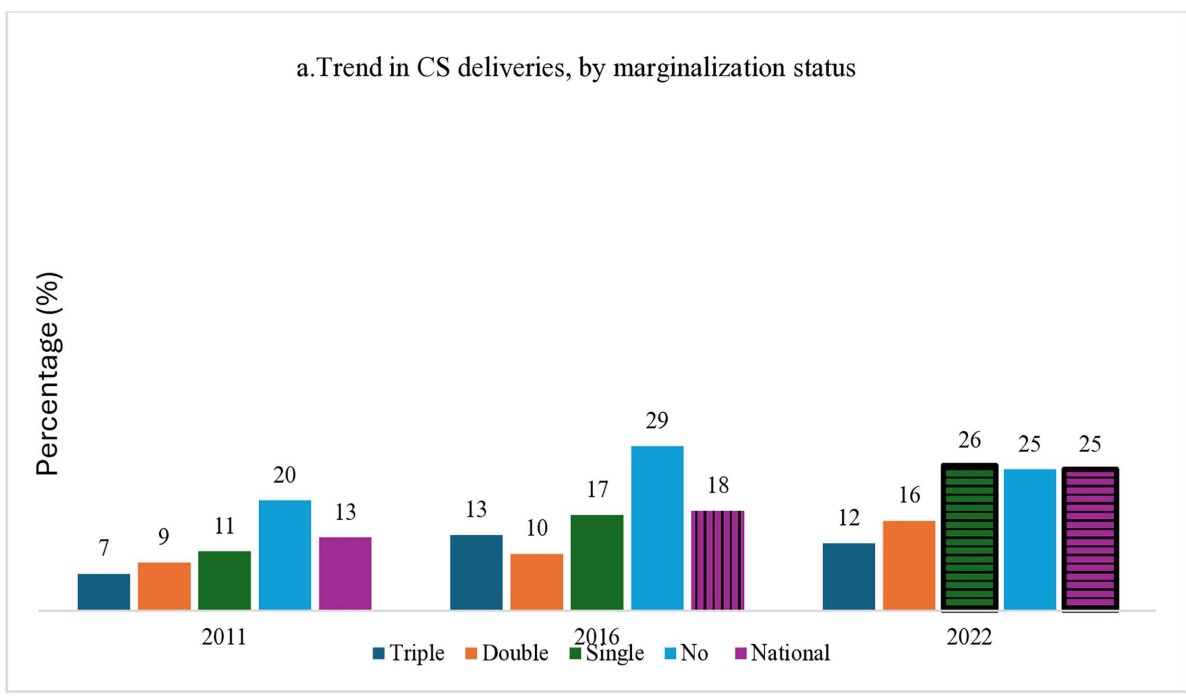

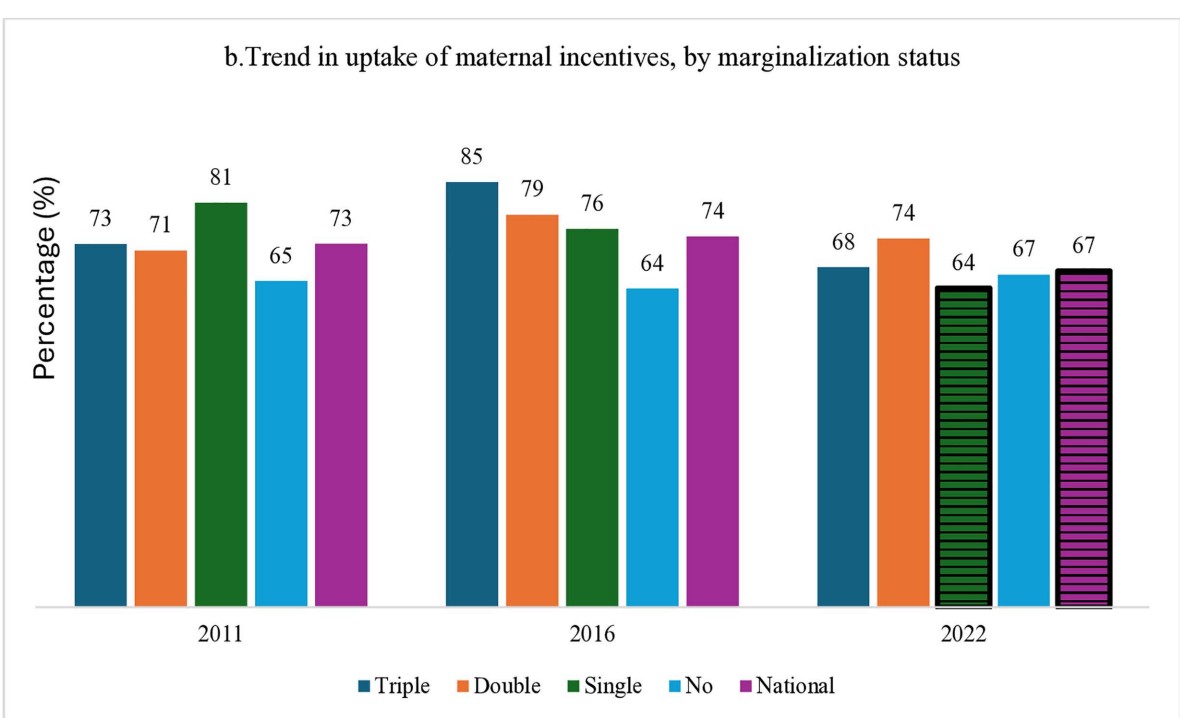

Note: Cut off p value for significance test (p<0.05). Each bar with vertical lines indicates a statistically significant change from 2011 to 2016. Each bar with horizontal lines indicates a statistically significant change from 2016 to 2022. Each bar with a solid outline indicates a statistically significant change from 2011 to 2022.

**Fig 4. Trends of delivery by CS and uptake of maternity incentive among women who had at least a live birth and gave birth in HFs one year prior to the surveys.**

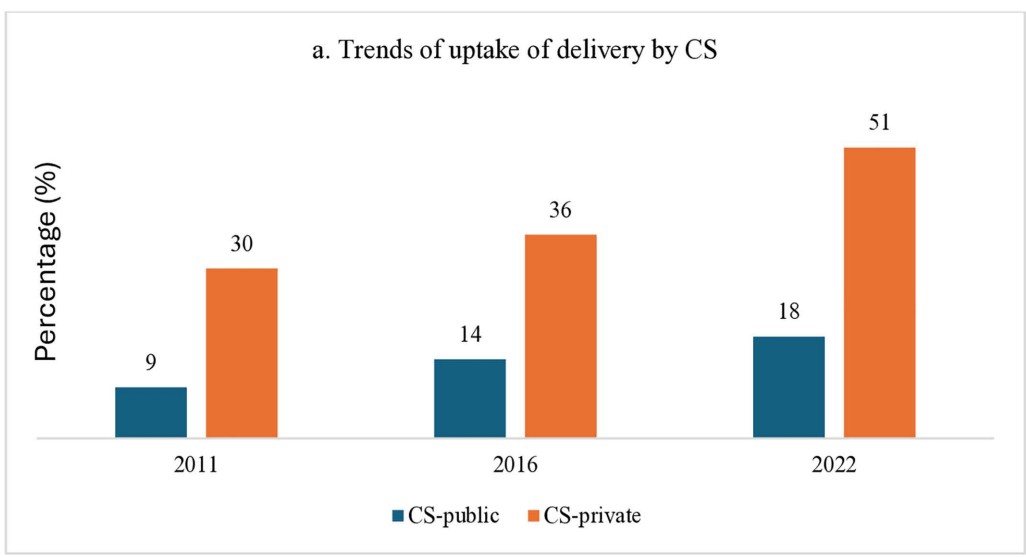

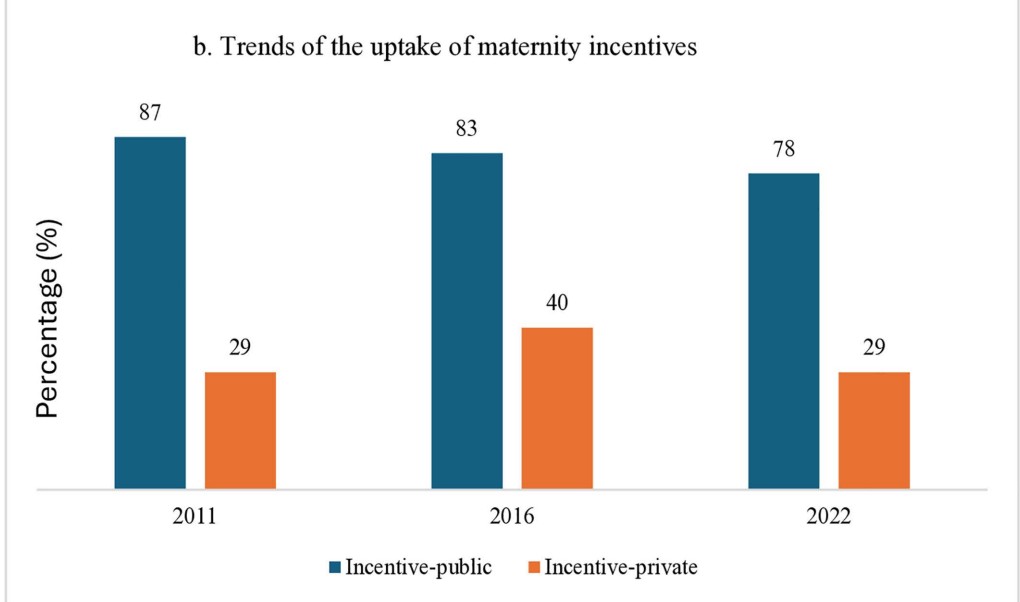

**Fig 5. Trends of uptake of delivery by CS and trends of the uptake of maternity incentives among women who had at least a live birth and gave birth in HFs one year prior to the three most recent surveys.**

Among all institutional delivery (n = 796), childbirth by CS per background variables, and their bivariable association (chi-square association) were included in S1 Appendix. Among all institutional deliveries, one in four women (25%) had delivery by CS. Compared with the national average and the other categories for each variable, delivery by CS was more common among women aged 35 and older (39.5%), women with higher education (42.6%), women in the highest wealth quintile (41.6%), women with no disadvantages (38%), women in Bagmati province (37.9%), women with paid jobs (39.3%), and women with a second birth order (33.1%).

Among all institutional delivery (n = 796), uptake of maternity incentive by background variables of study participants, and their bivariable association (chi-square association) were included in S1 Appendix. Two-thirds (67.3%) received

**Table 1. Distribution of women aged 15–49 years who had a live birth one year prior to the survey by background variables, NDHS 2022.**

| Maternal characteristics | % | 95% CI | N = 981 |
|---|---|---|---|
| **Structural factors** | | | |
| **Religion (N = 981)** | | | |
| Hindu | 83.7 | 79.8–87.0 | 822 |
| Other (e.g., Buddhism, Islam) | 16.3 | 13.0–20.2 | 159 |
| **Ethnicity(N = 981)** | | | |
| Brahmin | 8.2 | 6.4–10.8 | 81 |
| Chhetri | 18.7 | 15.6–22.1 | 183 |
| Madheshi | 17.0 | 13.7–21.0 | 167 |
| Dalit | 17.2 | 13.9–21.1 | 169 |
| Janajati | 31.9 | 27.8–36.3 | 313 |
| Muslim | 7.0 | 4.4–10.9 | 68 |
| **Education(N = 981)** | | | |
| No education | 16.4 | 13.6–19.7 | 161 |
| Basic | 33.4 | 30.2–36.8 | 328 |
| Secondary | 44.9 | 40.8–49.1 | 441 |
| Higher | 5.3 | 3.3–8.1 | 51 |
| **Wealth quintile (N = 981)** | | | |
| Poorest | 20.4 | 17.4–23.7 | 200 |
| Poorer | 21.6 | 18.6–25.1 | 212 |
| Middle | 20.1 | 17.1–23.5 | 197 |
| Richer | 20.2 | 17.0–23.7 | 198 |
| Richest | 17.7 | 14.4–21.6 | 174 |
| **Occupation(N = 981)** | | | |
| Not working | 44.9 | 41.1–48.7 | 440 |
| Agriculture | 40.6 | 36.6–44.7 | 398 |
| Manual labor | 5.8 | 4.3–7.9 | 57 |
| Working paid | 8.7 | 6.9–11.0 | 86 |
| **Marginalization status(N = 981)** | | | |
| Triple | 8.4 | 6.5–10.6 | 82 |
| Double | 29.8 | 26.3–33.7 | 293 |
| Single | 43.4 | 39.5–47.4 | 426 |
| No | 18.4 | 15.0–22.4 | 180 |
| **Intermediary factors** | | | |
| **Province(N = 981)** | | | |
| Koshi | 19.2 | 16.3–22.4 | 188 |
| Madhesh | 24.4 | 21.3–27.7 | 239 |
| Bagmati | 16.3 | 13.2–19.9 | 160 |
| Gandaki | 6.7 | 5.1–8.7 | 66 |
| Lumbini | 15.9 | 13.8–18.2 | 156 |
| Karnali | 7.1 | 6.1–8.2 | 70 |
| Sudurpashchim | 10.4 | 9.0–12.0 | 102 |
| **Residence(N = 981)** | | | |
| Urban | 66.0 | 62.8–69.2 | 648 |
| Rural | 34.0 | 30.8–37.2 | 333 |
| **Ecoregion(N = 981)** | | | |
| Mountain | 6.3 | 3.9–9.5 | 60 |

*(Continued)*

**Table 1.** (Continued)

| Maternal characteristics | % | 95% CI | N = 981 |
|---|---|---|---|
| Hill | 34.9 | 30.0–40.0 | 342 |
| Terai | 59.0 | 53.8–64 | 579 |
| **Native language(N = 981)** | | | |
| Nepali | 48.5 | 43.9–53.2 | 476 |
| Maithili | 18.8 | 14.5–23.9 | 185 |
| Bhojpuri | 8.5 | 5.5–12.8 | 83 |
| Other (e.g., Newari, Tharu, Tamang) | 24.2 | 20.6–28.2 | 237 |
| **Age in years (N = 981)** | | | |
| <20 | 18.8 | 16.4–21.6 | 185 |
| 20–24 | 36.8 | 33.8–39.9 | 361 |
| 25–29 | 27.6 | 24.6–30.9 | 271 |
| 30–34 | 12.7 | 10.2–15.9 | 125 |
| 35 and above | 4.1 | 2.8–5.6 | 39 |
| **Birth order (N = 981)** | | | |
| 1 | 40.8 | 36.9–44.6 | 400 |
| 2 | 36.8 | 33.3–40.5 | 361 |
| 3+ | 22.4 | 19.5–25.7 | 220 |
| **Place of delivery(N = 981)** | | | |
| Public HF | 63.5 | 59.3–67.6 | 624 |
| Private HF | 17.6 | 14.3–21.4 | 172 |
| Home | 18.9 | 16.0–22.2 | 185 |

maternity incentives. Compared with the national average and respective reference categories for each variable, uptake of maternity incentives was higher if women were of Chhettri ethnicity (73.8%), from the lowest wealth quintile (79.3%), from Gandaki province (90.4%), from the Mountain ecoregion (83.5%), or spoke other languages (74.1%).

## Determinants of institutional delivery in NDHS 2022

The crude odds ratio (COR) for background variables associated with delivery in HF are presented in S1 Appendix. Table 2 presents the adjusted odds ratio (AOR) and their 95% confidence intervals (CI) for various background characteristics associated with institutional delivery in 2022 NDHS. Lower odds of institutional delivery were observed for women under 20 years old (AOR = 0.50; 95% CI:0.28–0.88) and those from Karnali (AOR = 0.34; 95% CI:0.15–0.76) when compared to women aged 20–24 years and residing in Sudurpashchim [Table 2]. Similarly, lower odds of institutional delivery were found among women with triple (AOR = 0.29; 95% CI:0.09–0.9) and double (AOR = 0.29; 95% CI:0.11–0.79) forms of disadvantages as compared to women with no disadvantage. Institutional delivery was lower among women working in the agricultural sector (AOR = 0.61; 95% CI:0.39–0.97) as compared to those not working. Furthermore, institutional delivery was lower among women who are native Maithili speakers (AOR = 0.38; 95% CI:0.16–0.91), Bhojpuri speakers (AOR = 0.35; 95% CI:0.13–0.92) and other language speakers (AOR = 0.41; 95% CI:0.22–0.77), as compared to women who are native Nepali speakers. Compared to women having their first birth, those with a second birth had lower odds of institutional delivery (AOR = 0.24; 95% CI:0.13–0.43), while women with a third of higher birth order had even lower odds (AOR = 0.16; 95% CI:0.08–0.30). On the other hand, higher odds of institutional delivery were found if women had completed at least four ANC visits (AOR = 1.78; 95% CI:1.18–2.68).

**Table 2. Determinants of institutional delivery for women who had a live birth one year before the survey, NDHS 2022.**

| Maternal characteristics | Categories | AOR | 95% CI |
|---|---|---|---|
| **Structural factors** | | | |
| **Occupation** | Not working | 1.00 | |
| | Agriculture | 0.61* | 0.39–0.97 |
| | Manual labor | 0.82 | 0.32–2.08 |
| | Working paid | 1.11 | 0.39–3.20 |
| **Religion** | Hindu | 1.00 | |
| | Other (e.g., Buddhism, Islam) | 1.04 | 0.59–1.84 |
| **Multiple marginalization** | Triple | 0.29* | 0.09–0.90 |
| | Double | 0.29* | 0.11–0.79 |
| | Single | 0.51 | 0.18–1.40 |
| | No | 1.00 | |
| **Intermediary factors** | | | |
| **Province** | Koshi | 0.66 | 0.32–1.38 |
| | Madhesh | 0.42 | 0.16–1.10 |
| | Bagmati | 1.11 | 0.41–3.01 |
| | Gandaki | 0.63 | 0.25–1.59 |
| | Lumbini | 0.65 | 0.27–1.54 |
| | Karnali | 0.34** | 0.15–0.76 |
| | Sudurpashchim | 1.00 | |
| **Residence** | Urban | 1.00 | |
| | Rural | 0.92 | 0.61–1.40 |
| **Native language** | Nepali | 1.00 | |
| | Maithili | 0.38* | 0.16–0.91 |
| | Bhojpuri | 0.35* | 0.13–0.92 |
| | Other (e.g., Newari, Tharu, Tamang) | 0.41** | 0.22–0.77 |
| **Age** | <20 | 0.50* | 0.28–0.88 |
| | 20–24 | 1.00 | |
| | 25–29 | 0.98 | 0.60–1.60 |
| | 30–34 | 1.47 | 0.71–3.04 |
| | 35 and above | 1.26 | 0.52–3.02 |
| **Birth order** | First | 1.00 | |
| | Second | 0.24*** | 0.13–0.43 |
| | Third or Higher | 0.16*** | 0.08–0.30 |
| **Health system factors** | | | |
| **At least four ANC visits** | No | 1.00 | |
| | Yes | 1.78** | 1.18–2.68 |

Significant at * p<0.05, ** p<0.01, *** p<0.001.

## Determinants of institutional delivery in private HFs in NDHS 2022

The CORs for background variables associated with delivery in private HF in the 1 year prior to the 2022 NDHS are presented in S1 Appendix. Table 3 demonstrates the determinants of live childbirth in private HF. As compared to Sudur-pashchim, the odds of delivery in private HF were higher if women were from Koshi (AOR = 6.53; 95% CI:2.71–15.74), Madhesh (AOR = 4.69; 95% CI:1.6–13.75), Bagmati (AOR = 2.71; 95% CI:1.02–7.19), and Lumbini (AOR = 2.69; 95%

CI:1.13–6.4) provinces. Furthermore, women with a single form of disadvantage (AOR = 3.68; 95% CI:1.40–9.63) and no forms of disadvantages (AOR = 3.7; 95% CI:1.18–11.6) had higher odds of delivery in private HF compared to those with triple forms of disadvantages. On the other hand, there were lower odds of childbirth in private HF if women were from Karnali province (AOR = 0.08; 95% CI:0.01–0.63), working in manual labor (AOR = 0.35; 95% CI:0.13–0.94), and birth order three or more (AOR = 0.43; 95% CI:0.19–0.95) compared to women residing in Sudurpashchim province, women without any work, and women with first birth order [Table 3].

Table 3. Determinants of childbirth in private HF among women who had a live birth one year before the survey, NDHS 2022.

| Maternal characteristics | Categories | AOR | 95% CI |
|---|---|---|---|
| **Structural factors** | | | |
| **Occupation** | Not working | 1.00 | |
| | Agriculture | 0.95 | 0.58–1.55 |
| | Manual labor | 0.35* | 0.13–0.94 |
| | Working paid | 1.23 | 0.55–2.76 |
| **Religion** | Hindu | 1.00 | |
| | Other (e.g., Buddhism, Islam) | 0.99 | 0.56–1.75 |
| **Multiple marginalization** | Triple | 1.00 | |
| | Double | 1.76 | 0.64–4.88 |
| | Single | 3.68** | 1.40–9.63 |
| | No | 3.70* | 1.18–11.6 |
| **Intermediary factors** | | | |
| **Province** | Koshi | 6.53*** | 2.71–15.74 |
| | Madhesh | 4.69** | 1.60–13.75 |
| | Bagmati | 2.71* | 1.02–7.19 |
| | Gandaki | 0.91 | 0.28–2.97 |
| | Lumbini | 2.69* | 1.13–6.40 |
| | Karnali | 0.08* | 0.01–0.63 |
| | Sudurpashchim | 1.00 | |
| **Residence** | Urban | 1.00 | |
| | Rural | 0.97 | 0.59–1.59 |
| **Native language** | Nepali | 1.00 | |
| | Maithili | 2.10 | 0.83–5.34 |
| | Bhojpuri | 0.78 | 0.25–2.44 |
| | Other (e.g., Newari, Tharu, Tamang) | 0.87 | 0.49–1.54 |
| **Age** | <20 | 1.00 | |
| | 20–24 | 0.82 | 0.43–1.58 |
| | 25–29 | 1.15 | 0.54–2.47 |
| | 30–34 | 2.12 | 0.82–5.50 |
| | 35 and above | 0.76 | 0.21–2.75 |
| **Birth order** | First | 1.00 | |
| | Second | 0.69 | 0.40–1.19 |
| | Third or higher | 0.43* | 0.19–0.95 |
| **Health system factors** | | | |
| **At least four ANC visits** | No | 1.00 | |
| | Yes | 0.98 | 0.52–1.88 |

Significant at * p < 0.05, ** p < 0.01, *** p < 0.001.

## Determinants of delivery by CS in NDHS 2022

The CORs for background variables associated with delivery by CS in the one year prior to the 2022 NDHS are presented in S1 Appendix. Table 4 demonstrates the determinants of CS among women who had live births one year before the survey from NDHS 2022. The study found higher odds of delivery by CS if women were aged 25–29 years (AOR = 4.42; 95% CI:2.05–9.54), 30–34 years (AOR = 4.57; 95% CI:1.84–11.31), and 35 years and above (AOR = 5.67; 95% CI:1.7–18.94)

**Table 4. Determinant of delivery by CS for women who had a live birth in HF one year prior to the survey, NDHS 2022.**

| Maternal characteristics | Categories | AOR | 95% CI |
|---|---|---|---|
| **Structural factors** | | | |
| **Occupation** | Not working | 1.00 | |
| | Agriculture | 0.48** | 0.30–0.79 |
| | Manual labor | 0.64 | 0.24–1.75 |
| | Working paid | 0.87 | 0.43–1.78 |
| **Religion** | Hindu | 1.00 | |
| | Other (e.g., Buddhism, Islam) | 0.96 | 0.50–1.84 |
| **Multiple marginalization** | Triple | 1.00 | |
| | Double | 1.62 | 0.54–4.87 |
| | Single | 2.14 | 0.77–5.91 |
| | No | 1.83 | 0.64–5.2 |
| **Intermediary factors** | | | |
| **Province** | Koshi | 4.53*** | 2.26–9.11 |
| | Madhesh | 2.76* | 1.17–6.50 |
| | Bagmati | 3.45** | 1.59–7.52 |
| | Gandaki | 1.88 | 0.76–4.66 |
| | Lumbini | 2.49* | 1.19–5.22 |
| | Karnali | 0.91 | 0.33–2.51 |
| | Sudurpashchim | 1.00 | |
| **Residence** | Urban | 1.00 | |
| | Rural | 0.71 | 0.46–1.10 |
| **Native language** | Nepali | 1.91* | 1.04–3.50 |
| | Maithili | 2.13* | 1.05–4.32 |
| | Bhojpuri | 0.82 | 0.24–2.87 |
| | Other (e.g., Newari, Tharu, Tamang) | 1.00 | |
| **Age** | <20 | 1.00 | |
| | 20–24 | 1.99 | 0.99–3.99 |
| | 25–29 | 4.42*** | 2.05–9.54 |
| | 30–34 | 4.57** | 1.84–11.31 |
| | 35 and above | 5.67** | 1.7–18.94 |
| **Birth order** | First | 1.00 | |
| | Second | 1.04 | 0.64–1.69 |
| | Third or Higher | 0.4* | 0.19–0.84 |
| **Health system factors** | | | |
| **At least four ANC visits** | No | 1.00 | |
| | Yes | 1.11 | 0.55–2.26 |

Significant at * p<0.05, ** p<0.01, *** p<0.001.

compared to women who were teenage mothers. Similarly, this study found higher odds of delivery by CS if women were from Koshi (AOR = 4.53; 95% CI:2.26–9.11), Madhesh (AOR = 2.76; 95% CI:1.17–6.5), Bagmati (AOR = 3.45; 95% CI:1.59–7.52), and Lumbini (AOR = 2.49; 95% CI:1.19–5.22) provinces compared to women from Sudurpashchim province. Higher delivery by CS was also found among native Nepali speakers (AOR = 1.91; 95% CI:1.04–3.5) and native Maithili speakers (AOR = 2.13; 95% CI:1.05–4.32) compared to women who speak other languages. Lower odds of delivery by CS were found among women working in the agricultural sector (AOR = 0.48; 95% CI:0.30–0.79) compared with nonworking women. Additionally, women with a birth order of three or more had lower (AOR = 0.4; 95% CI:0.19–0.84) odds of delivery by CS compared to women with a first birth order [Table 4].

### Determinants of the uptake of maternity incentives in NDHS 2022

The CORs for background variables associated with the uptake of maternity incentives in the one year prior to the 2022 NDHS are presented in S1 Appendix. Table 5 demonstrates the determinants of uptake of maternal incentives among women in NDHS 2022. The analysis found lower odds of uptake of maternity incentives among women from Koshi (AOR = 0.11, 95% CI:0.05–0.25), Madhesh (AOR = 0.18, 95% CI:0.13–0.7), Bagmati (AOR = 0.3, 95% CI:0.13–0.7), and Lumbini (AOR = 0.33, 95% CI:0.16–0.71) provinces compared to Sudurpashchim province.

### Discussion

This study reveals significant inequities across various demographics, indicating lower completion rates and high discontinuation along the maternity continuum, particularly among women facing multiple forms of disadvantage compared to their advantaged counterparts in Nepal. It highlights disparities in institutional delivery rates, with lower rates observed among women from Karnali and Madhesh provinces, native speakers of Terai languages (Maithili and Bhojpuri), and those with higher birth orders. Conversely, the completion of at least four ANC visits is associated with higher rates of institutional delivery, underscoring the importance of early and consistent maternal care. The trend toward CS has increased in the private HFs, especially among women with fewer disadvantages and older age groups. In contrast, women working as labourers and those with higher birth orders were less likely to utilize private health services. The utilization of private health services was notably higher in the Koshi, Bagmati, Madhesh, and Lumbini provinces. Additionally, the uptake of maternity incentives was significantly lower among those who opted for private health services and underwent delivery by CS (not all private HFs implement maternity incentive program). These findings underscore critical inequities in maternal health, emphasizing the necessity for targeted interventions and a layered approach that addresses the trends and key determinants affecting access to maternal care for disadvantaged women.

The completion rates of all three routine maternal care visits were low, with a higher uptake of ANC observed in early gestational months, followed by a significant decline in the later weeks of pregnancy. Difficulties in reaching HF as women near childbirth, coupled with a lack of transportation and insufficient availability of uninterrupted care across the continuum, could have contributed to the imbalance between catch-up and keep-up rates [39]. The combination of high institutional delivery and low PNC reveals that many mothers who gave birth in HFs but did not receive PNC. This suggests that socially excluded women, including those from remote or marginalized communities, face greater challenges such as lack of nearby facilities and unreliable transport and intensified by low family support and financial constraints, which discourage them from seeking timely maternal care [40]. These findings were consistent with the further analysis of NDHS 2016 [41] and similar studies conducted in other LMICs, such as Zambia [42] and Pakistan [43]. The higher rates of first ANC visit, followed by drops during the late gestational age and childbirth, could be attributed to physical barriers to accessing HFs, particularly due to distance and inadequate transportation facilities. For instance, most HFs in Nepal are in the center of municipalities, which might be inaccessible for some remote communities under their catchment areas, compounded by shortage of skilled health workforce for maternal care, facility preparedness including shortage of essential medicine, and the lack of functional referral system [44]. Improving transportation facilities could enhance access to HFs during

**Table 5. Determinants of uptake of maternity incentives by women who had a live birth one year prior to the survey, NDHS 2022.**

| Maternal characteristics | Categories | AOR | 95% CI |
|---|---|---|---|
| **Structural factors** | | | |
| **Occupation** | Not working | 1.00 | |
| | Agriculture | 1.08 | 0.71–1.64 |
| | Manual labor | 1.65 | 0.75–3.63 |
| | Working paid | 0.8 | 0.45–1.42 |
| **Religion** | Hindu | 1.00 | |
| | Other (e.g., Buddhism, Islam) | 1.75 | 0.94–3.26 |
| **Multiple marginalization** | Triple | 1.00 | |
| | Double | 1.03 | 0.45–2.35 |
| | Single | 0.58 | 0.26–1.31 |
| | No | 0.74 | 0.29–1.91 |
| **Intermediary factors** | | | |
| **Province** | Koshi | 0.11*** | 0.05–0.25 |
| | Madhesh | 0.18*** | 0.07–0.45 |
| | Bagmati | 0.3** | 0.13–0.70 |
| | Gandaki | 1.54 | 0.49–4.82 |
| | Lumbini | 0.33** | 0.16–0.71 |
| | Karnali | 0.86 | 0.40–1.86 |
| | Sudurpashchim | 1.00 | |
| **Residence** | Urban | 1.00 | |
| | Rural | 1.09 | 0.71–1.66 |
| **Native language** | Nepali | 0.75 | 0.44–1.28 |
| | Maithili | 0.51 | 0.24–1.06 |
| | Bhojpuri | 0.65 | 0.24–1.73 |
| | Other (e.g., Newari, Tharu, Tamang) | 1.00 | |
| **Age** | <20 | 1.00 | |
| | 20–24 | 1.20 | 0.71–2.04 |
| | 25–29 | 1.22 | 0.65–2.30 |
| | 30–34 | 0.69 | 0.29–1.66 |
| | 35 and above | 1.05 | 0.36–3.05 |
| **Birth order** | First | 1.00 | |
| | Second | 1.46 | 0.91–2.34 |
| | Third or higher | 1.61 | 0.84–3.09 |
| **Health system factors** | | | |
| **At least four ANC visits** | No | 1.00 | |
| | Yes | 1.37 | 0.81–2.29 |

Significant at ** p<0.01, *** p<0.001.

the gestational weeks around childbirth [45,46]. Another strategy could be establishing waiting homes for women from remote areas to minimize the long travel time for women nearing childbirth. Evidence suggests maternal waiting homes could improve institutional delivery towards better perinatal health outcomes [47]. Ensuring adherence to quality of care guidelines could facilitate the continuum of care from ANC and PNC, and mandating PNC before discharge from HFs after delivery could prove beneficial [41,48].

Women with multiple forms of marginalization had lower odds of institutional delivery such as women with low wealth status and lack of education, those residing in remote provinces like Karnali, and women with higher birth orders. Marginalized women generally face structural inequities (poverty, unemployment), have difficulty with living conditions [49], and could face several access barriers to reach HFs [50], including difficult geographic terrains and distance to HF. Despite the implementation of the Aama program that was designed to address financial barriers, expand birthing centers to address geographic issues, and train SBAs to improve the access and quality of childbirth services in birthing centers. However, evidence suggests that care quality at birthing centers needs to improve for improved maternal health outcomes [23,50]. Native speakers of regional language, such as Maithili and Bhojpuri languages from Terai region, had lower odds of delivery in HFs compared to native Nepali speakers. The cultural upbringings and belief systems related to care-seeking practices and the sharing of reproductive and maternal health issues could have influenced the disparities. The language barriers between the Nepali-speaking care providers and those who cannot speak Nepali may hinder effective communication of health needs, potentially widening equity gaps. Health care providers who share similar ethnic, cultural, and linguistic backgrounds with local women have a better opportunity to effectively address their maternal health issues [51].

There was an increasing trend of delivery in private HFs with higher odds of childbirth in provinces with better socioeconomic development indicators (e.g., Koshi and Bagmati, and provinces that cover the Terai region, such as Madhesh and Lumbini), and among relatively privileged women with single or no marginalization. In principle, private health services complement UHC as their role is vital, whereas the public health system is inadequate in providing health services to its population. However, in the context of the poorly regulated private HFs and the lack or inadequate coverage of a health insurance program (national health insurance program), private health services can financially burden people and push them to further marginalization. Studies revealed bypassing local HFs to go to bigger hospitals and public hospitals in nearby cities caused overcrowding that further compromised the quality of maternal care in Nepal [52,53]. People sought private care to receive better quality of care, despite having to pay for those services [53]. Private HFs that are profit motivated often intend on over-prescription of unnecessary tests, medications, and surgeries, which also increase the burden of care cost, especially for socioeconomically disadvantaged groups [54,55]. Nepal's private health services are fee-for-service, and people must pay for health services. GON has implemented the Aama program in all public HFs, but very few private HFs have implemented this program [56]. However, the care-seeking patterns for childbirth in private HFs are increasing. Seeking care in private HFs means paying for services that are freely available in public HFs [54].

High delivery by CS was found if women were pregnant at an older age and from Koshi, Bagmati, Lumbini, or Madhesh provinces and among Maithili native-speaking women. The WHO has recommended that 10–15% of childbirth should require delivery by CS [57]. However, Nepal has a higher rate of CS than this recommendation and has three times higher CS rate if women were in private HFs [58]. Among women who had multiple forms of marginalization (poor, illiterate, and without education), more than two-thirds (72%) had childbirth in private HFs by CS, and women from Koshi, Bagmati, Lumbini, and Madhesh provinces. Madhesh province has lower socioeconomic indicators, including female literacy and distinct linguistic and cultural diversity, which influence the ability to reach HFs for services, especially in pregnancy and childbirth [59]. Higher CS rates in private HFs may be linked with revenue-generation mechanisms rather than being medically recommended. This illuminates existing policy gaps, such as weak regulation of the private health sectors, inadequate insurance coverage, and ineffective accreditation systems. The NDHS dataset does not capture the medical indication for CS delivery (e.g., elective vs. emergency), limiting the interpretation of whether these increased CS rates reflect genuine obstetric need or are driven by other factors such as provider preference, financial incentives, or patient demand. Medicalization and commercialization of routine care services can have high-cost implications for those socioeconomically disadvantaged populations. HFs get higher unit cost reimbursement for CS through maternity incentive program (for each CS, HFs receive NPR 10,000 which approximately 90 USD). The monitoring system needs to be strengthened to ensure the CS services are provided only when medically indicated.

Overall, less than two-thirds of all women who gave birth in HFs received maternity incentives nationally, and the trend is decreasing. Though maternity incentive interventions are more equitable than other interventions, they are not universally utilized in Nepal. There are several potential reasons behind not achieving universal coverage maternity incentive program in Nepal. Our findings (reducing equity but not universal) were similar to maternity incentive programs in Bangladesh (Maternal Health Voucher Scheme) [60], and India (Pradhan Mantri Matru Vandana Yojana) [61]. Provinces with higher rates of childbirth in private HFs and higher rates of delivery by CS had lower odds of uptake of maternity incentives. The proportion of women with multiple forms of marginalization who received maternity incentives was higher than their privileged counterparts. Nearly four in five women received incentives if they delivered babies in public HFs, while the uptake of incentives was notably decreasing among women who gave birth in private HFs. This could be because most public HFs have implemented the Aama program, while not all private HFs have adopted it. This could also have contributed to the lower uptake rates in private HFs than public HFs [27]. Families continue to make out-of-pocket expenses for birth in private HFs, despite receiving maternity incentives in public hospitals [62]. Concerns about quality of care, high delivery in the private sector, and high CS delivery costs in private HFs, along with the lack of uniformity of maternity incentives program between private and public HFs should be reevaluated to align with the changing context of the decentralized health system in Nepal.

## Study limitations

The study included the mothers who had live births one year prior to the surveys. The study used this short window of time to reduce the recall bias of childbirth-related information and identify the recent history of maternal care. However, this short window of time has the implications of a small sample size in some of the categories of variables. Therefore, we merged some categories of variables (e.g., marginalization status) with a small sample size. Additionally, we created disadvantages using three variables, such as education, wealth status, and ethnicity which were multicollinear with the multiple marginalization status. Thus, we excluded these three variables in the regression analysis. The analysis of the uptake of maternity incentives in private HFs should be interpreted carefully since the denominator includes all private HFs, regardless of whether they are implementing the maternity incentive scheme. In addition, further limitations of this study include cross sectional design, self-reported data, and no quality-of-care indicator in the analysis. Self-reported data may cause recall bias for ANC/PNC timing. We limited our sample for those women who had at least one live birth one year prior to the survey to make short window of recall period. There are data limitations such as the NDHS data collected information related to incentive where few private HFs implement Aama program, and there is no cross verification of CS whether that was elective or medically indicated for the CS. In the future, NDHS could include such questions to reduce the data limitations.

## Implications for policy, practice and research

Some implications for policy, practice and research can be drawn from this study. First, policies should focus on delivering health services to the most marginalized groups by addressing structural and intersectional marginalization, improving conditions where they live, and enhancing access to quality healthcare through multisectoral collaboration. Second, to boost completion rates for routine maternal care visits, the maternal incentive program should be integrated with continuity of care indicators and tracked in the health management information system. Policies should also ensure that women receive at least one PNC visit within 24 hours of childbirth for mothers and newborns to secure incentives. Third, healthcare providers need to follow mandatory PNC for mothers and newborns for mothers delivering in HFs, supported by regular training and quality monitoring. Improved communication during ANC visits about childbirth locations is also essential to improve health literacy and client satisfaction. Fourth, recruiting local healthcare providers can enhance communication and care in provinces like Madhesh and Lumbini, which have high populations of Maithili- and Bhojpuri-speaking women. Additionally, improving transportation in Karnali province is crucial to increasing access to HFs for childbirth. Fifth, ensuring uninterrupted access to quality childbirth services in HFs can enhance trust in public HFs and health services. Local healthcare providers can better identify women facing multiple disadvantages and tailor service delivery strategies

accordingly. It's essential to regulate the private sector by enforcing guidelines, such as allocating 10% of beds for marginalized populations and standardizing care costs. The government should also promote the Aama program in private HFs to create a supportive environment. As care-seeking in private HFs rises due to a lack of trust in public facilities, improving quality in public HFs, particularly in remote areas and Madhesh province, is crucial. The Aama program should be effectively implemented in both public and private settings. Finally, the rising rate of CS in the private HFs is concerning, as it's unclear if these procedures are medically necessary or requested by patients, or hidden interest of profit-motived care providers or HFs. The WHO estimates, only 5–15% of total deliveries require CS as our findings exceeds this estimates that means women in Nepal opt elective surgeries, but reasons could be different. This could be one of the important research questions to further explore, focusing on why Nepal has high CS rate, especially in private HFs. Furthermore, effective health information and client education on the benefits and risks of normal delivery and CS are essential. Maternity incentive programs should be revised to specifically target women of multiple marginalization, such as poor women who belong to disadvantaged ethnicities and those who live in remote areas. Engaging local leaders and stakeholders in understanding the impact of intersecting marginalization can strengthen health policy efforts at both local and provincial levels. Broadly, policy and programmatic recommendations can be made for structural and systemic reforms to address the non-health sector factors (e.g., improved socioeconomic status, and development approach for road networks); program-level reforms (e.g., revised incentives, mandatory PNC, quality audits); and regulatory/monitoring reforms (e.g., CS audits, regulation of private HFs, linking incentives to equity metrics).

## Conclusions

This study finds that while first ANC visits and institutional births are high, PNC visits and completion of all three maternal care visits are low, with significant equity gaps. Women who are facing multiple forms of marginalization, and those from rural province (e.g., Karnali) have lower access to maternal care compared to more privileged groups. There is a notable rise in childbirth and delivery by CS in private HFs, particularly among Maithili- and Bhojpuri-speaking women and those from specific provinces, but low uptake of maternity incentives. The increasing trend of deliveries in private HFs and deliveries by CS, combined with the lack of maternity incentives could lead to financial hardship while seeking health care. It is essential to design and implement focused efforts to improve completion rates from ANC to PNC, better data integration into health monitoring systems, and regular training for care providers to enhance service quality. The Aama program should be expanded to cover the full continuum of care uniformly across facilities. Addressing supply-side barriers, recruiting local health workers, and improving public trust in health systems are essential, especially in remote communities. To achieve equity and universality of maternal services in Nepal, it is imperative to implement interventions at multiple levels including improving road network; hiring local health workers and training on cultural competence; revision of maternity incentive and its implementation in private HFs; creatioon and inclusion of index of all three care visits in routine health information management system; quality of care audits; and regulation of private HFs and audits of CS.

## Supporting information

**S1 Appendix. Supporting information (description of study variables, descriptive analysis of the dependent variables with all outcome variables, descriptive output of trend analysis for all outcome variables, output of bivariable analysis, and crude odds ratios of all outcome variables with independent variables).**
(DOCX)

## Acknowledgments

The authors would like to thank the DHS program for granting permission to use the data for further analysis. The authors acknowledge the inputs and guidance for this entire work from USAID Nepal, Ministry of Health and Population Nepal, and ICF, USA.

## Author contributions

**Conceptualization:** Resham B Khatri, Sabita Tuladhar.

**Data curation:** Resham B Khatri.

**Formal analysis:** Resham B Khatri.

**Investigation:** Resham B Khatri, Rolina Dhital.

**Methodology:** Resham B Khatri.

**Project administration:** Resham B Khatri, Sabita Tuladhar.

**Resources:** Resham B Khatri.

**Software:** Resham B Khatri.

**Supervision:** Yibeltal Assefa.

**Validation:** Resham B Khatri, Rolina Dhital, Sabita Tuladhar, Nisha Joshi Bhatta, Yibeltal Assefa.

**Visualization:** Resham B Khatri, Rolina Dhital, Sabita Tuladhar, Nisha Joshi Bhatta, Yibeltal Assefa.

**Writing – original draft:** Resham B Khatri.

**Writing – review & editing:** Resham B Khatri, Rolina Dhital, Sabita Tuladhar, Nisha Joshi Bhatta, Yibeltal Assefa.

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
