## [Decision Letter · Decision Letter 0]

5 Sep 2025

Dear Dr. Khatri,

Thank you for submitting your manuscript to PLOS ONE. After careful consideration, we feel that it has merit but does not fully meet PLOS ONE’s publication criteria as it currently stands. Therefore, we invite you to submit a revised version of the manuscript that addresses the points raised during the review process.

We look forward to receiving your revised manuscript.

Kind regards,

Kanchan Thapa, MPH, MPhil

Academic Editor

PLOS ONE

Journal Requirements:

2. We note that your Data Availability Statement is currently as follows: All relevant data are within the manuscript and in Supporting Information files.

3. Please upload a copy of Figure 10, to which you refer in your text on page 13. If the figure is no longer to be included as part of the submission please remove all reference to it within the text.

Additional Editor Comments:

Dear Authors,

Please work on abstract section to highlight the importance of the paper for global reader, low and middle income countries and why such study is important.

In background section of the manuscript, same information should be added. The paper only highlight the national importance. I suggest to add more information about global, regional and epecially LMICS country context.

In method section, how did you work with sample weight for three rounds of the survey? Please indicate it.

Line 196- You mention something about stata command. How does it address all the statistical calculation for 3 rounds of the survey?

Did you use any checklist such as STROBE for the present analysis? If yes, where it is? If not why? Please includes information for all the three rounds of study?

Total sample= Year I, Year II, Year III

Excluded= Year I, Year II, Year III

Included =

Sample= weighted vs non-weighted in analysis?

Ethics section,

Please include how the ethical approval was obtained. In the latest survey, there was covid 19 how it was addressed? Please cite the relevant DHS report indicating ethics information.

Results

Table 1. provides only information about 2022 survey, is it correct? And how about N or n here?

Table 2-5, you mention about AOR for different determinants. Among these, which are the most powerful factors? And how the impacts of these factors changed over the time? For an example, the impact of education, family size, husband occupations were most important factors in 2012 (let’s say model I). How does it changed over the time in 2022. Are the factors are the same?

Discussion

You stated about the significant inequalities among the groups. Is it based on all the data set of DHS or only based on 2022 information.

Major comments: You paper profoundly stated about equity gap, but nowhere in method section you stated about any test of equity. Please write about the equity test you performed and indicate relevant figure based on the equity test such as Lorenz Curve or others?

Data analysis section:

You indicate about VIF less than or equal to 5, what is the basis for such inclusion. You can present the VIF test results as supplementary file which will guide further researcher how to look at VIF results.

Did you perform any factorial analysis? What is the basis of performing or not performing such analysis?

You have included a lot of information for analysis, among them which are the most important factors, how is its role changed over time?

Figure 1. does it represent your study flow? What is the theoretical basis for such a conceptual framework creation such as any health belief model or any models? Why people come to utilize health services?

Figure 2. is congested, you can mention about % in y axis and remove % in each data text inside figure.

You can rework to make clear about the data reported in each figures. Was the figure based on weighted average or not? Do you want to show CI for each information?

Uniformity in figure design is missing. Please include y- axis as percentage and make the figure less congested.

Figure 6. your y-axis title is missing.

Reviewers' comments:

Reviewer's Responses to Questions

**Comments to the Author**

1. Is the manuscript technically sound, and do the data support the conclusions?

Reviewer #1: Yes

Reviewer #2: Yes

Reviewer #3: Yes

Reviewer #4: Yes

Reviewer #5: Yes

Reviewer #6: Yes

Reviewer #7: Yes

Reviewer #8: Yes

Reviewer #9: Yes

Reviewer #10: Yes

2. Has the statistical analysis been performed appropriately and rigorously?

Reviewer #1: Yes

Reviewer #2: Yes

Reviewer #3: Yes

Reviewer #4: Yes

Reviewer #5: Yes

Reviewer #6: No

Reviewer #7: Yes

Reviewer #8: Yes

Reviewer #9: Yes

Reviewer #10: Yes

3. Have the authors made all data underlying the findings in their manuscript fully available?

Reviewer #1: Yes

Reviewer #2: Yes

Reviewer #3: Yes

Reviewer #4: Yes

Reviewer #5: Yes

Reviewer #6: Yes

Reviewer #7: Yes

Reviewer #8: Yes

Reviewer #9: Yes

Reviewer #10: Yes

4. Is the manuscript presented in an intelligible fashion and written in standard English?

Reviewer #1: Yes

Reviewer #2: Yes

Reviewer #3: Yes

Reviewer #4: Yes

Reviewer #5: Yes

Reviewer #6: Yes

Reviewer #7: Yes

Reviewer #8: Yes

Reviewer #9: Yes

Reviewer #10: Yes

Reviewer #1: This paper presents a strong, equity-focused analysis of maternal health service utilization trends in Nepal using three NDHS datasets, with an innovative intersectional disadvantage index that adds policy relevance. To further strengthen the manuscript, the authors should deepen theoretical framing (e.g., Three Delays Model), clarify data limitations, enhance discussion with more analytical depth and global comparisons, report statistical significance of trends, and expand on actionable policy implications tailored to Nepal’s federal health system. Please see the attached paper for my suggestions.

Reviewer #2: Thank you for the opportunity to review this manuscript. The topic is very important and highly relevant at this time. However, there are several areas for improvement:

Overall: The authors should avoid common patterns found in AI-generated text and consider redrafting the manuscript in more natural, human language. There are instances of long dash lines (-) appearing in many places, as well as an overuse of phrases such as "underscores" and "highlights."

Here are some of my suggestions.

Abstract:

1. years is missing in the sentence "....among 25 women aged 15–49 who had experienced at least one live birth prior to each survey"

2. Please include some key numbers in the results section.

Introduction

1. 1. The introduction section appears to be too lengthy; it would be better to shorten it.

2. You have talked about socioeconomic status here. Please also talk about the availability, accessibility, and equity gaps in maternal health services in Nepal.

3. There are several terms such as "marginalization," "disadvantage," and "multiple marginalization." Please reconsider these terms and ensure consistency throughout.

Methods:

1. Please include details about the sample size, such as how DHS sampling was conducted. Additionally, specify the type of multivariate analysis performed, such as logistic regression. The tables also need footnotes to clarify the variables adjusted for in the multivariate models.

Results:

1. There are other categories in many tables, better to add what are included in that category.

Discussion:

1. The section on private sector deliveries highlights issues related to “profit motives and unnecessary procedures,” which can help illuminate existing policy gaps, such as weak regulation, inadequate insurance coverage, and ineffective accreditation systems.

2. While the Aama program is mentioned, it is important to address whether it effectively reduces inequities and to explore why its reach in private facilities is limited.

3. Additionally, it would be beneficial to include information about systemic gaps in the continuum of care, such as transportation, health workforce capacity, facility preparedness, and the referral system.

Reviewer #3: (1) As the current manuscript is prepared using the data of three years, however, Determinants of institutional delivery were assessed through multivariable analysis of the NDHS 2022 dataset. Why the data from remaining two years (2011 and 2016) were not utilized for this outcome?

(2) After the completion of all procedures we prepare a manuscript, hence, preparing the final manuscript in the past tense is suitable.

(3) "Analyses" or "Analysis", once go through this as per requirement.

(4) Arrange the key words in the alphabetical orders.

(5) Line 163: "were categories into", is the tense correct?

(6) "(dis)advantages", this word is little bit uneasy to understand by common viewer.

(7) Line 193: "variation inflation factor (VIF)", is it variation or variance?

(8) Line 278, Table 1: N = 981, however, total number of male and female is 982, and same is found in other columns as well, do you have any specific reason please?

(9) Province "Sudurpaschim" has been considered as reference group, if the rational behind it is described it would be easy to readers.

(10) Regarding age group, both the "<20 years" and "20-24" years are taken as reference groups, was there any specific description?

(11) Digits after decimal are expected to be uniform, however, some findings are described with single digit after decimal.

(12) Line 29 says you have applied multivariable analysis, Line 189 says you applied multivariate analyses, did you apply both the methods? As they are different methods.

Reviewer #4: This was a secondary data analysis to reflect the maternal health services in Nepal and provides insightful evidence on this sector. Improving maternal health is a national priority in Nepal, supported by the government’s Aama suraksha program, which provides free delivery care and financial incentives to encourage women to access the maternal health care. Using nationally representative survey data (Nepal Demographic Health Survey of different years), this study examined whether women are receiving the recommended continuum of maternal health services-antenatal care (ANC), delivery in a health facility, and postnatal care (PNC)-and whether there are differences across social and geographic groups.

This study found that most women attend their first ANC visit and deliver in health facilities, reflecting progress in service use. However, far fewer women receive timely PNC, and only a small proportion complete all three essential services. Gaps are especially large among women from disadvantaged groups, rural communities, and those speaking Maithili or Bhojpuri. Private facilities are increasingly used for childbirth, including cesarean deliveries, but maternal incentive uptake remains low in these settings. The findings suggest the need for stronger efforts to improve continuity of care, expand incentives, and address barriers faced by disadvantaged women to ensure more equitable maternal health outcomes in Nepal.

The authors have aroused needy health gaps in low resources settings like Nepal, and have presented the Nationally represented data for more generalizability. However, this study needs some revisions to give audience insightful and non-misguided interpretations. The review comments are attached too.

Reviewer #5: The authors have submitted a technically sound article using the latest data of NDHS. The statistical analysis was done correctly but in some cases few changes can make the results more meaningful. In this manuscript as the figures cant be added, there was some lack of clarity regarding the conceptual frameworks and few more figures. However, The article is helpful for policymakers to plan for the related activities in future. It can surely be published with minor changes.

Reviewer #6: • Comment 1 (Background section):

The statement “Nonetheless, the maternal mortality ratio (MMR) remains high” is vague. Instead of presenting it hypothetically, the authors should provide clear evidence with specific data or figures to support the statements.

• Comment 2 (Introduction section):

The authors mentioned that “Over the past few decades, Nepal witnessed an increase in household income, wealth status, life expectancy, access to education, and basic health services.” However, to my knowledge, this Annual Report of Nepal, does not measure household income and wealth status. The authors should cite appropriate sources such as the Nepal Living Standards Survey or surveys conducted annually by Nepal Rastra Bank.

• Comment 3 (Data analysis section):

The authors reported that they checked the Variance Inflation Factor (VIF) for multicollinearity and excluded some independent variables. It would improve clarity if the authors specified which variables exhibited collinearity and were subsequently removed from the regression models.

• Comment 4 (Ethical section):

The statement “the survey received ethical approval from the ICF Institutional Review Board in the USA and the Nepal Health Research Council, Nepal” is unclear. The authors should write the full form of ICF to avoid confusion, as it could refer to different organizations.

• Comment 5 (Analysis and Results section):

The authors mentioned in statistical analysis section, adjusted odds ratios (AORs) were calculated in bivariate analysis using the Chi-Square test. This is methodologically incorrect because the Chi-Square test does not account for confounders and therefore cannot produce an AOR, whilst only limit on crude (unadjusted) odds ratio (CORs). Logistic regression should be used instead.

The authors should also clarify the criteria used for selecting variables into multivariate analysis. In addition, the table headings currently use the generic term “determinants of…”; it would be more precise to state the exact type of analysis performed.

Reviewer #7: The authors have well prepared the manuscript, which is technically sound. The methodological part is well explained according to the NDHS report. Few queries and suggestions regarding variables and typo error has been provided in the attached file below.

Reviewer #8: The manuscript is well structured and written in standard English. It follows the guideline of PLOS one. The author have used the nationally representative data sets which strengthens the validity of the findings in the study. However there are some minor grammatical errors which needs to be corrected to improve the flow. Some sentences are too long and needs to be shortened.

Reviewer #9: 1. Inconsistent Sample Sizes

The manuscript's "Methods" and "Results" sections present conflicting information regarding the sample size used for the determinant analysis. The abstract and "Methods" section state that the determinant analysis for institutional delivery was performed on the 2022 NDHS dataset. However, the "Methods" section specifies that the total sample size for the determinant analysis of institutional delivery, delivery by CS, and maternal incentives was 796 women who gave birth at a health facility. In contrast, the "Results" section states that the descriptive analysis of institutional delivery was based on 981 study participants. This discrepancy in sample sizes between the methodology and the results section for the same analysis is unclear.

2. Unclear Explanation of the "Disadvantage Status" Variable

The manuscript explains that a new variable, "disadvantage status," was created by combining three background variables: education, wealth status, and ethnicity. It details the process of dichotomizing these variables and then merging the categories. The final variable is said to have four categories: triple, double, single, and no disadvantages. However, the description of how the categories with "at least one form of disadvantage" and "two forms of disadvantage" were merged is confusing and does not clearly explain which specific combinations of the original eight categories were merged to create the final four. This lack of clarity makes it difficult to replicate the analysis.

Reviewer #10: Overall Impression of the Paper:

The paper looks well structured and has provided a glimpse of health inequity and its effects in one of the most important areas, maternal service utilization.

Strengths of the Paper:

Wise use of NDHS data that represent the whole nation

Has provided a good rationale and aim

Areas to improve:

Abstract:

The abstract looks quite lengthy, consider it concise.

Methods:

In the Methods section (line number 151-153) under data sources, the sample size and weighting could be explained quite a little more to make the readers unfamiliar with this concept clearer.

In the study variables under methods, section, explanation of outcomes and exposures could be done in separate paragraphs under these headings to make the picture clear.

Provide reference for advantaged ethnicities.

Creation of marginalization status and disadvantaged status could be better explained using a flowchart.

Figures are yet to be inserted, so it makes the concept cloudy at places.

Results:

The results are well presented but contain many tables, consider merging the tables where relevant. Also highlight the major findings in tables so that the navigation gets easier. If possible, show 2-3 results in a single graph using multiple colors to indicate different findings’ trend and provide indices.

Discussion:

Line number 404 and 405: Terminologies such as catch up rate and keep up rate need to be defined. Also, define what are intersectional groups.

Discussion is done in different paragraphs for different findings which makes things clear and reading easier but considering the length of all the paragraphs can be done to make it concise.

Policy implications are written in a separate paragraph, well-done. Adding recommendations on indications for CS procedure monitoring specially in private health facilities and also providing incentives under Aama Program even in private HFs with reference can be valuable.

In the limitations section, adding the caution for interpretation on a small subgroup (Muslim Dalit, Mountainous regions) can be done.

Conclusion is well-written, consider it breaking down to two paragraphs: Conclusion and Recommendations

**Do you want your identity to be public for this peer review?** For information about this choice, including consent withdrawal, please see our Privacy Policy

Reviewer #1: No

Reviewer #2: No

Reviewer #3: No

Reviewer #4: No

Reviewer #5: **Yes: ** Binika Shrestha

Reviewer #6: No

Reviewer #7: **Yes: ** Vijaya Laxmi Shrestha

Reviewer #8: No

Reviewer #9: No

Reviewer #10: No

---

## [Author Response · Author response to Decision Letter 1]

11 Oct 2025

Comments from the editor

Comment: Please work on abstract section to highlight the importance of the paper for global readers, low- and middle-income countries and why such study is important.

Response: some global and regional context is added to draw the attention of global readers.

Comment: In background section of the manuscript, some information should be added. The paper only highlights national importance. I suggest adding more information about global, regional and especially LMICS country context.

Response: Thanks for the global and regional context of maternal health has added in the introduction section.

Comment: In method section, how did you work with sample weight for three rounds of the survey? Please indicate it.

Response: Weighted sample was identified using the sampling weight given in the individual survey for specific year. There is no need to adjust the sampling weight of pooled data of three surveys. This has been indicated in the revised manuscript. Please look for detailed analysis code in DHS-Analysis-Code/Analysis_Reports/FA152_maternalhealthequity/FA152_MaternalHealth_71824.do at main · DHSProgram/DHS-Analysis-Code · GitHub

Comment: Line 196- You mention something about Stata command. How does it address all the statistical calculations for 3 rounds of the survey?

Response: The descriptive analysis of individual survey was calculated and presented the trends of each outcome variable. Their significance was tested using z-test for proportion taking data from individual survey using two sample significance difference of proportion (proportion test). While multivariable logistic analysis was conducted for the data of the NDHS 2022 only. Please look for code used in this analysis ; DHS-Analysis-Code/Analysis_Reports/FA152_maternalhealthequity/FA152_MaternalHealth_71824.do at main · DHSProgram/DHS-Analysis-Code · GitHub

Comment: Did you use any checklist such as STROBE for the present analysis? If yes, where is it? If not, why? Please include information for all three rounds of study.

Response: This study was based on the further analysis of publicly available secondary data. Yes, we followed most of the steps of STROBE checklist to report the findings and draft this manuscript.

Comment: Total sample= Year I, Year II, Year III, excluded= Year I, Year II, Year III,

Included =Sample= weighted vs non-weighted in analysis?

Response: As mentioned above, we used the secondary data of NDHSs, we briefly mentioned that how NDHS designed sampling procedure and how final respondents are selected for the interview, and how information is collected. Detailed sampling size and methods is described in the original reports of each NDHS, we have referenced them for further readings for interested readers. More specifically, NDHS collected information related to maternal health care utilization among mothers who had live birth five years prior to the survey. But in our analysis, we included women who had live birth one year prior to each survey. We have mentioned that all sample included in the analysis are weighted otherwise specifically indicated.

Comment: Ethics Section-Please include how ethical approval was obtained. In the latest survey, there was COVID-19 how it was addressed. Please cite the relevant DHS report indicating ethics information.

Response: this is further analysis of secondary data derived from the nationally representative surveys in Nepal – i.e. Nepal Demographic and Health Surveys (2011, 2016, and 2022). Ethical approval for those surveys was described in their original reports which we have cited in the methods section of our manuscript. In our manuscript, we have briefly described on how ethical procedure occurred in the NDHS survey. For this analysis, the researcher team was granted approval from ICF to download and use data for the further analysis.

Comment: Results-Table 1. provides only information about 2022 survey, is it correct? And how about N or n here?

Response: table 1 provides the distribution of sample included in the study. Descriptive analysis of the sample is always calculated as column percentage. In our analysis, for NDHS 2022, N was 981 for all variables, and numbers of sample for each variable categories (n) are reported in each row of each variable. In the revised table, N is included in each study variable as well.

Comment: Table 2-5, you mention about AOR for different determinants. Among these, which are the most powerful factors? And how have the impacts of these factors changed over time? For example, the impact of education, family size, husband occupations were most important factors in 2012 (let’s say model I). How does it change over the time in 2022. Are the factors the same?

Response: the analysis output included in tables 2-5 represents the outputs of binomial logistic regression analysis. Our study objective was to analyse the trends of maternal health service utilization over three surveys at the national level and by marginalization status. We examined the determinants of maternal health service analysis using data of NDHS 2022 only. We did not investigate the trends of change in associated determinants over time or most powerful determinants impacting the maternal service utilization as this was not our study objective.

Comment: Discussion-You stated about the significant inequalities among the groups. Is it based on all the data set of DHS or only based on 2022 information.

Response: Our study objective was to identify the associated determinants of maternal health service utilization using data of NDHS 2022.

Comment: Major comments: Your paper profoundly stated about equity gap, but nowhere in method section you stated about any test of equity. Please write about the equity test you performed and indicate relevant figure based on the equity test such as Lorenz Curve or others?

Response: we investigated the uptake of the maternal service utilization among women with multiple forms of marginalization (composite intersectional variable created using wealth status, ethnicity and education status- see our published paper based on NDHS2016, https://doi.org/10.1186/s12889-021-11142-8). In the current analysis, detailed of this approach is described in methods section and reported finding. Our equity analysis approach was different than conventional equity analysis (e.g. Lorenz curve or concentration index).

Comment: Data analysis section: You indicate VIF less than or equal to 5, what is the basis for such inclusion. You can present the VIF test results as supplementary file which will guide further researchers on how to look at VIF results.

Response: All Stata codes including calculation of VIF used to analyse this paper are publicly available in the GitHub webpage where researchers can access there. Here is the link of GitHub: DHS-Analysis-Code/Analysis_Reports/FA152_maternalhealthequity/FA152_MaternalHealth_71824.do at main · DHSProgram/DHS-Analysis-Code · GitHub

Comment: Did you perform any factorial analysis? What is the basis of performing or not performing such analysis? You have included a lot of information for analysis, among them which are the most important factors, how has its role changed over time?

Response: As described in response to one of the comments above, our study objective did not require conducting factorial analysis. So, we did not conduct factorial analysis. Our study objective to look the descriptive trends of maternal service utilization in the last three survey (overall, and by marginalization status), and associated determinants of maternal health service utilization in NDHS 2022. There are several studies have already available which examined the determinants of maternal health services utilization data of previous NDHSs i.e. NDHSS 2011, or 2016.

Comment: Figure 1. does it represent your study flow? What is the theoretical basis for such a conceptual framework creation such as any health belief model or any models? Why do people come to utilize health services?

Response: Figure 1, a conceptual framework (which is different than analytical framework) is a thinking framework to guide our analysis. Our study is further analysis of secondary data and identify trends of maternal health services utilization among women in Nepal. As social determinants of health framework conceptualize, the underpinning idea behind health equity are founded on the structural, intermediary, and health system factors. Health equity depends on the structural inequity (such as education, ethnicities or wealth status) which are generally non-modifiable by health systems or non-health sector interventions (developmental approaches). Thus, to address the structural factors towards health equity, political and structural reforms are needed such as addressing poverty, ethnic inequalities. Health systems can provide health service to poor but cannot reduce poverty or improve wealth status. Intermediary factors are those factors which can be modified by development approaches (such as improving living, working, and travel conditions by making bridges or road etc). In our analysis, we consider geography or province as intermediary factors, if we improve road networks or bridge in the river, then the distance to health facility can be reduced that improve the access health services. Health system factors that can address through health systems including service readiness, quality of care etc (Detailed in Towards equity of maternal and newborn health services in Nepal - UQ eSpace). The figure 1, i.e., conceptual framework represents those underpinning ideas of social determinants of health frameworks, and previous publications including the publication of first author of this paper. We used conceptual framework to guide our analysis based on the available variables and data in the NDHS 2022 dataset.

Comment: Figure 2. is congested, you can mention about % in y axis and remove % in each data text inside figure.

Response: figures have been revised and edited as suggested.

Comment: You can rework to make clear the data reported in each figure. Was the figure based on weighted average or not? Do you want to show CI for each piece of information?

Response: All data were weighted otherwise indicated. Figures will become more congested if we include the CI. So, we have kept data level as it is.

Comment: Uniformity in figure design is missing. Please include y- axis as percentage and make the figure less congested.

Response: figures have been revised and edited as suggested.

Comment: Figure 6. Your y-axis title is missing.

Response: figure have been revised and edited as suggested.

Reviewer #1:

Comment: This paper presents a strong, equity-focused analysis of maternal health service utilization trends in Nepal using three NDHS datasets, with an innovative intersectional disadvantage index that adds policy relevance. To further strengthen the manuscript, the authors should deepen theoretical framing (e.g., Three Delays Model), clarify data limitations, enhance discussion with more analytical depth and global comparisons, report statistical significance of trends, and expand on actionable policy implications tailored to Nepal’s federal health system.

Please see the attached paper for my suggestions.

Response: we thank reviewer for these insightful and border feedback. As this study used health equity framework as conceptual framework and framed the research to answer who are benefited from the service utilization and who are left behind. For this we used adapted and modified version of social determinants of health and frame paper accordingly. The model like Three Delays Model is also important to describe when and where maternal deaths occur, and such framework can be useful to describe the more grounded research using primary data, more specifically qualitative data. Our research question was to analysis the secondary data and answer the question such as who are seeking maternal health service and national level and group level. For which we took secondary data, and we conducted further analysis. If our research question was like why women are not seeking care, what factors are contributing at household or community level, why women are not reaching health facilities, or what factors are influencing while seeking care in hospitals, then models like Three Delays Model could be more appropriate model. However, in our analysis, the second component of Three Delays Model (Delays in reaching health facilities or health workers for maternal care) was interlinked in our analysis. We have added some statement why women are seeking care (due to geographical factors or due to multiple marginalization status), we have incorporated those arguments in our interpretation and discussion.

Comment: Abstract - The result lacks quantitative depth. Consider stating the percentage point increase in institutional delivery or CS. Clarify whether the decrease in maternity incentive use is statistically significant.

Response: Qualitative information was supported by quantitative information in the abstract. Only significant information was included in the abstract.

Comment: Introduction -No global comparison of trends in institutional delivery, CS, or incentive programs (e.g., how does Nepal compare to other LMICs?)

Response: We have included brief global and regional context of maternal health in the beginning of the introduction section.

Comment: Results -No statistical test results are reported for trend changes. Provide p-values for trend comparisons across years.

Response: output of the significance test results was reported in the figure. We have included the supplementary files and detailed results output including p values.

Comment: Findings on CS trends are important, but causes (e.g., medical indication vs. elective) are not discussed—this may lead to misinterpretation of high CS rates. So better to add the line like- “However, the NDHS dataset does not capture the medical indication for caesarean delivery (e.g., elective vs. emergency), limiting the interpretation of whether these increased CS rates reflect genuine obstetric need or are driven by other factors such as provider preference, financial incentives, or patient demand.” Please discuss this point in the discussion section as well.

Response: Thanks for the suggestions. The suggested statements have included and discussed in the discussion section. We have incorporated the causes of the CS. As per WHO estimates, only 5-15% of total deliveries require CS as our findings exceeds this estimates that means women in Nepal opt elective surgeries, but reasons could be different why they opt for that. This is one of the research questions for further exploration which have discussed in our paper (study implication section). There are data limitations such as the NDHS data collected information related to incentive where few private HFs implement Aama program, and there is no cross verification of CS whether that was elective or medically indicated for the CS. In the future, NDHS could include such questions to reduce the data limitations.

Comment: Discussion -This reads like a repetition of results. The first paragraph should instead summarize the meaning of findings, not just restate them.

Response: we revised the discussion section, and remove the repetitive content with the result section, and interpreted findings. Our framing of the discussion section, first paragraph with summary of key findings. As we have multiple outcome variables under the trends analysis and determinants of four outcome variables related to maternal health service utilization. In the subsequent paragraphs we interpreted findings on each key outcome variables and structured each paragraph taking key variables under study and compared and contract study findings with studies from other similar settings/ countries. We have included some subsections on strengths and limitations of the study, and implications of the study findings.

Comment: You should strengthen your discussion by connecting your findings to an established theoretical framework—in this case, the Three Delays Model—to better explain why women drop out of the maternal care continuum (e.g., from ANC to institutional delivery to PNC)

---

## [Decision Letter · Decision Letter 1]

10 Nov 2025

Unpacking trends and gaps in Nepal’s progress on maternal health service utilization: insights from the most recent Demographic and Health Surveys (2011, 2016 and 2022)

PONE-D-25-36549R1

Dear Dr. Khatri,

We’re pleased to inform you that your manuscript has been judged scientifically suitable for publication and will be formally accepted for publication once it meets all outstanding technical requirements. At the mean time, I suggest you to address all the minor comments raised by reviewer and also address the typos. 

Kind regards,

Kanchan Thapa, MPH, MPhil

Academic Editor

PLOS ONE

Additional Editor Comments (optional):

Dear Authors,

I suggest you to once review the comments raised by peer reviewer and address all the typos. Please provide updated information as raised by reviewer.

Reviewers' comments:

Reviewer's Responses to Questions

**Comments to the Author**

Reviewer #1: All comments have been addressed

Reviewer #2: All comments have been addressed

Reviewer #3: All comments have been addressed

Reviewer #6: All comments have been addressed

Reviewer #7: All comments have been addressed

Reviewer #9: All comments have been addressed

2. Is the manuscript technically sound, and do the data support the conclusions?

Reviewer #1: Yes

Reviewer #2: Yes

Reviewer #3: Yes

Reviewer #6: Yes

Reviewer #7: Yes

Reviewer #9: Yes

3. Has the statistical analysis been performed appropriately and rigorously?

Reviewer #1: Yes

Reviewer #2: Yes

Reviewer #3: Yes

Reviewer #6: Yes

Reviewer #7: Yes

Reviewer #9: Yes

4. Have the authors made all data underlying the findings in their manuscript fully available?

Reviewer #1: Yes

Reviewer #2: Yes

Reviewer #3: Yes

Reviewer #6: Yes

Reviewer #7: Yes

Reviewer #9: Yes

5. Is the manuscript presented in an intelligible fashion and written in standard English?

Reviewer #1: Yes

Reviewer #2: Yes

Reviewer #3: Yes

Reviewer #6: Yes

Reviewer #7: Yes

Reviewer #9: Yes

Reviewer #1: Thank you for nicely addressing all the comments and suggestions. I have accepted this manuscript for the publication.

Reviewer #2: Authors have addresses my suggestions. The manuscript now has been improved. I do not have further suggestions. Thank you for genereting this evidence that is very important for LMICs.

Reviewer #3: All the comments have been answered in proper and systematic ways. Similarly, the manuscript is technically sound and data are also informative and they address the conclusion. In this manuscript statistical methodology is also properly applied as per the requirements. As all the data are available and clear to understand. Along with these, the authors have prepared manuscript in standard English which is easy to understand by all the readers.

Reviewer #6: The revised version shows a marked improvement over the previous draft, reflecting the author’s thoughtful incorporation of feedback and diligent revisions. The manuscript is now well-structured and substantially refined. I sincerely appreciate the author’s efforts, and I have no further comments at this stage.

Reviewer #7: The authors have well prepared the manuscript and incorporated few suggestions given earlier. There are still few questions which are unanswered.

Abstract: How can you conclude this statement in Line 54-56 (……..worsens financial burden)? I don’t see the same arguments in Conclusion of the study. Please make sure it is not overstated.

Please explain how did you assess the PNC visit within 48 hours. Government has protocol of 24 hours, 3rd day and 7th day. Did you combine 1st and 2nd PNC visits?

Since, there is no provision of maternal incentives providing by all private facilities, is it fair to make a comparison on rise or decline of up taking incentives in private and public facilities?

There has been a revision on providing maternal incentives such as Rs 1000 in Terai, 2000 in Hills and 3000 in mountain. So, it would be nice to state the updated provision too in introduction part Line 146-150.

Some of the results are found repeatedly stated in discussion section. Avoid redundancy. Repetition of same thing may not be interesting to readers. Try to present your findings in a simpler way.

The authors have analyzed determinants of maternal service utilization individually in multiple levels; however, I couldn’t find if the determinants of continuum of maternal care were analyzed. Since the complete uptake was found only 59% so it would be great to know what were the factors associated with it.

In line 621-623, We limited our sample for those women who had at least one live birth one year prior to the survey to reduce the survey. Is it correct?

The conclusion and recommendations are well written. Please check again for typo error.

Reviewer #9: (No Response)

**Do you want your identity to be public for this peer review?** For information about this choice, including consent withdrawal, please see our Privacy Policy

Reviewer #1: No

Reviewer #2: No

Reviewer #3: No

Reviewer #6: No

Reviewer #7: No

Reviewer #9: No

---

## [Editor Report · Acceptance letter]

PONE-D-25-36549R1

PLOS ONE

Dear Dr. Khatri,

I'm pleased to inform you that your manuscript has been deemed suitable for publication in PLOS ONE. Congratulations! Your manuscript is now being handed over to our production team.

Kind regards,

on behalf of

Mr. Kanchan Thapa

Academic Editor

PLOS ONE